



**The summertime Saharan heat low: Sensitivity of the radiation budget and atmospheric heating to water vapor and dust aerosol.**

Netsanet K. Alamirew[1*], Martin C. Todd[1*], Claire L. Ryder[2], John M. Marsham[3], Yi Wang[1]

[1]Department of Geography, University of Sussex, Brighton, UK

[2]Department of Meteorology, University of Reading, Reading, UK

[3]School of Earth and Environment, University of Leeds, Leeds, UK

[*]Correspondence to: N. Alamirew na286@sussex.ac.uk or M. Todd m.todd@sussex.ac.uk

**Abstract.** The Saharan heat low (SHL) is a key component of the West African climate system and an important driver of the West African Monsoon across a range of timescales of variability. The physical mechanisms driving the variability in the SHL remain uncertain, although water vapour has been implicated as of primary importance. Here, we quantify the independent effects of variability in dust and water vapour on the radiation budget and atmospheric heating of the region using a radiative transfer model configured with observational input data from the Fennec field campaign at the location of Bordj Badji Mokhtar (BBM) in southern Algeria (0.9E, 21.4N), close to the SHL core, for June 2011. Overall, we find dust aerosol and water vapour to be of similar importance in driving variability in the top of atmosphere (TOA) radiation budget and therefore the column integrated heating over the SHL (~7 W m$^{-2}$ per standard deviation of dust AOD). As such we infer that SHL intensity is likely to be similarly enhanced by the effects of dust and water vapour surge events. However, the details of the processes differ. Dust generates substantial radiative cooling at the surface (~11 W m$^{-2}$ per standard deviation of dust AOD), presumably leading to reduced sensible heat flux into the boundary layer, which is more than compensated by direct radiative heating from SW absorption by dust in the dusty boundary layer. In contrast water vapour invokes a longwave radiative warming of at the surface of ~6 W m$^{-2}$ per standard deviation of column integrated water vapour in Kg m$^{-2}$. Net effects involve a pronounced net atmospheric radiative convergence with heating rates on average of 0.5 K day$^{-1}$ and up to 6 K day$^{-1}$ during synoptic/meso-scale dust events from monsoon surges and convective cold pool outflows ('haboobs'). On this basis we make inferences on the processes driving variability in the SHL associated with radiative and advective heating/cooling. Depending on the synoptic context over the region processes driving variability involve both independent effects of water vapour and dust and compensating events in which dust and water vapour are co-varying. Forecast models typically have biases of up to 2 kg m$^{-2}$ in column integrated water vapour (equivalent to a change in 2.6 W m$^{-2}$ TOA net flux) and typically lack variability in dust, and so are expected to poorly represent these couplings. An improved representation dust and water vapour and quantification of associated radiative impact is thus imperative in quest for the answer to what remains to be uncertain related with the climate system of the SHL region.

## 1. Introduction

During boreal summer the Saharan Heat Low (SHL), a low-level thermal low, extends over a vast sector of the central Sahara Desert, covering much of northern Mauritania, Mali and Niger and Southern Algeria (Figure 1). The area of low surface pressure is characterized by extremes of high surface temperature (Lavaysse et al.,





2009; Messager et al., 2010), and boundary layer (BL) depth (Marsham et al., 2013b), and is co-located with a global maximum in seasonal dust aerosol loading (Knippertz and Todd, 2012).

It is increasingly recognised that the SHL is a key component of the West African climate system and an important driver of the West African Monsoon across a range of timescales of variability e.g. (Chauvin et al., 2010; Couvreux et al., 2010; Lafore et al., 2010; Martin and Thorncroft, 2014; Martin et al., 2014; Parker et al., 2005; Peyrille and Lafore, 2007; Sultan and Janicot, 2003; Thorncroft and Blackburn, 1999; Xue et al., 2010). Notably, the intensification of the SHL in recent decades has been linked to the recovery of the Sahelian rainfall from the multi-decadal drought of the 1970s-90s, partly through a water vapour positive feedback process, in which radiative warming from increasing water vapour strengthens the SHL, which enhances the moist low level monsoon flow driving greater water vapour transport into the SHL and further warming (Dong and Sutton, 2015; Evan et al., 2015b) with an implied enhanced West African Monsoon.

The SHL results from a complex interplay of heating processes within the Saharan BL, from the conversion of large radiative surpluses at the surface into sensible heat flux, cooling from horizontal temperature advection, itself a function of the strength of the pressure gradient into the SHL core, and radiative cooling and adiabatic warming via subsidence in the upper BL (Alamirew et al., submitted). The SHL intensity is therefore likely to be modulated by radiative effects of variability in surface albedo, dust aerosol, water vapour and cloud which feedback onto the circulation and thus advective cooling, water vapour transport and the processes governing dust emission and transport. In addition, the SHL is also modulated by external dynamical controls on advective cooling from both tropical (Knippertz and Todd, 2012) and extra-tropical sources (Chauvin et al., 2010).

Previous studies have quantified direct radiative effects (DRE) of dust aerosol at the top of atmosphere (TOA) and surface from in situ observations and satellite data (Ansell et al., 2014; Banks and Brindley, 2013; Yang et al., 2009), whilst Marsham et al., 2016, hereafter M16, extend this empirically to consider water vapour variations, and implicitly cloud, as well as dust. However, there remain important gaps in our understanding. First, there are substantial uncertainties in the magnitudes of radiative fluxes (and other heat budget terms) across both the various reanalyses and observations. Second, separating the radiative effects of water vapour from both from both its and associated clouds and from dust aerosol is challenging from observations, given the strong co-variability of dust and total column water vapour (TCWV) anomalies in the Sahara associated with monsoon surges and resulting convective cold pool events ('haboobs') which transport water vapour and dust into the central Sahara (Garcia-Carreras et al., 2013; Marsham et al., 2008; Marsham et al., 2013b). As such, there is a need to quantify more fully the DRE of dust and water vapour, both independently and together, over the Sahara. This information is necessary to resolve the processes that govern the fundamental structure and maintenance and variability of the SHL. Addressing these research gaps is hindered by the acute shortage of routine observations in the region and large discrepancies between models and reanalyses (Evan et al., 2015a; Roberts et al., 2015).

This paper seeks to address these gaps in our understanding of radiative processes within the SHL during summer. Specifically, to quantify the separate roles of water vapour and dust aerosol in controlling the top of atmosphere, surface, and the vertical profile of the atmospheric column radiative budget. This will be achieved through radiative transfer (RT) model simulations using uniquely detailed observations of atmospheric conditions over the SHL region during summer, including those from the main supersite of the recent Fennec field campaign (Marsham et al., 2013b). Best estimates and associated uncertainty are established through a set of RT model experiments testing the sensitivity of radiative flux and atmospheric heating rates to variability in water vapour




and dust and to uncertainty in a set of controlling variables. In this way, we can inform interpretation of hypotheses on drivers of SHL variability and its wider impact on the regional climate.

## 2. Description of the Radiative Transfer model and observed radiative flux data

### 2.1 The SOCRATES Radiative Transfer model

The research questions are addressed through simulations from a column stand-alone RT model. Such models are commonly used for detailing the combined and unique radiative impact of dust aerosol and water vapour (Osborne et al., 2011; Osipov et al., 2015; Otto et al., 2007; Otto et al., 2009; Otto et al., 2011; Slingo et al., 2006). RT models typically comprise a radiative transfer core and a pre-processor to configure the necessary information on the radiatively active atmospheric constituents and surface characteristics. Typically, these include meteorological and gas component profiles from observations, reanalysis products or weather/climate models, spectral aerosol optical property profiles and surface properties.

Here we use the SOCRATES (Suite Of Community Radiative Transfer codes based on Edwards and Slingo) (Edwards and Slingo, 1996; Randles et al., 2013) model configured with observed and idealised profiles of water vapour and dust aerosol, as described below. SOCRATES is a flexible RT model, operated here in two streams of standalone radiative transfer code, which calculates the longwave and shortwave radiative fluxes and heating rates throughout the atmosphere given the atmospheric and surface properties of that column, the solar zenith for the location, date and time. Radiative flux calculations are made for parallel plane atmosphere with spectral resolution ranging over the shortwave and longwave from 0.2 to 10 μm and 3.3 μm to 10,000 μm, divided in 6 and 9 bands respectively. Inputs for the model are meteorological fields (temperature, specific humidity), active radiative gases ($O_2$, $CO_2$, $CH_4$, $CO$, $N_2O$, and $O_3$), surface fields (skin temperature, surface pressure, broadband albedo, and emissivity), dust aerosol optical properties and the vertical profile of dust mass mixing ratio. A detailed description of the model is given in (Randles et al., 2013).

We specify these inputs as accurately as possible using observations from the recent Fennec field campaign, which obtained unique data from within the SHL region during June 2011(Ryder et al., 2015). We use observations from ground-based instruments deployed at the main Fennec supersite at Bordji Badji Mokthar (BBM) 0.9E, 21.4N and 420m elevation close to the Aljeria-Mali border (Marsham et al., 2013b) and various aircraft flights (see Ryder et al., 2015 for overview) complemented, where direct observations are inadequate, with fields from the European Centre for Medium-Range Weather Forecasts (ECMWF) Interim Reanalysis (ERA-I) (Dee et al., 2011) and Modern-Era Retrospective analysis for Research and Application (MERRA) (Rienecker et al., 2011) reanalysis.

We undertake two types of RT experiment in this study (i) model 'configuration mode' in which we test the sensitivity of simulated radiative fluxes to uncertainty in as many of the input variables as possible, as described in Section 3.2, summarised in Table 1, with results provided in Section 4.1. This will result in an acceptably configured model for experimental analysis. (ii) Model 'experiment mode' to specifically address the research questions using the 'optimal' model configuration. The experiments are described in Section 3.3, summarised in Table 2, with results described in Section 4.2. These include RT model experiments run for durations from one day to one month (June 2011).





### 2.2 Observed top of atmosphere and surface radiation measurements for comparison with RT simulations

We use satellite retrievals of TOA radiation from two sources. 1). The EUMETSAT Geostationary Earth Radiation Budget (GERB) (Harries et al., 2005) level 2 products of Averaged Rectified Geolocated (ARG) fluxes at approximately 17 minute time and 50km spatial (at nadir) resolution, with spectral ranges 0.32 to 4μm in the shortwave and 4 to 100 μm in the longwave. 2). The Clouds and the Earth's Radiant energy System (CERES) (Wielicki, 1996) instrument which has channels that measure total radiance (0.4-200μm) and shortwave radiance (0.4-4.5μm). Since there is no longwave-only channel on CERES, daytime longwave radiances are determined

from the difference between the total and shortwave channel radiances. We use two CERES products: (i) the monthly mean Energy Balanced and Filled (EBAF) product at 1-degree resolution. (ii) The CERES Level-3 SSF1deg_Hour TERRA footprint gridded data (CERES-footprint) instantaneous, twice daily with 1-degree resolution.

For our high resolution, pixel based analysis focused on a single location (BBM), cloud screening is

15 notoriously challenging. For GERB data we apply the EUMETSAT cloud mask to derive clear sky and all-sky conditions and for CERES data we use both all sky and clear sky products. MODIS cloud parameters are used to derive CERES cloud free fluxes. However, analysis of GERB all-sky minus clear-sky fluxes at BBM suggests an unrealistically small cloud DRE (~2 W.m$^{-2}$ in longwave flux), which suggests that the cloud mask is not robust. We therefore choose only to use GERB all sky fluxes and limit the clear sky-only analysis to the CERES products.

For 'validation' of the 'optimum' model configuration, we favour comparison with GERB (all-sky) because the time period of the CERES monthly product is not exactly compatible with the RT simulations of 8-30[th] June, whilst the CERES footprint data has observations twice daily.

Surface measurements of shortwave and longwave upwelling and downwelling radiation are obtained from Kipp and Zonen CNR4 radiometers situated at 2m height deployed at BBM during the FENNEC campaign

(Marsham et al., 2013b). We also use the skin temperature product form CERES Level-3 SSF1deg_Hour TERRA footprint data.

### 3 Description of RT model experiments and input data

#### 3.1. Input data common to all experiments

#### 3.1.1 Atmospheric data

Profiles of temperature and water vapour are obtained from radiosonde measurements at BBM for June

8[th]-30[th] 2011 (Figure 2). The temporal resolution varied from 3-6 hourly over the study period. Profiles of trace gases needed by the radiative transfer model ($CO_2$, $O_2$, $N_2O$, $O_3$, and $CH_4$) are taken from the standard tropical atmosphere (Anderson et al., 1986). Temperature and water vapour profiles beyond the upper maximum height of the radiosonde data (approximately 20 km) are also taken from the standard tropical atmosphere. This is unlikely to affect RT model results significantly since there is little day to day variability in the uppermost part of

the atmosphere. A distinction can be made between the cooler, drier, less dusty Saharan 'maritime' phase from




around 8[th] to 12[th] June to a hotter, moister, dustier 'heat low' phase from around 13[th] to 30[th] June (Figure 2a) during which time both synoptic scale monsoon surges and meso-scale convective cold pool events transported both water vapour and dust into the heart of the SHL (see Ryder et al., 2015; Todd et al., 2013 for full details). For comparison, profiles of water vapour from radiosonde measurements with Era-I reanalysis are shown in Figure

5    2b. Despite the good agreement between measurement and model outputs, ERA-I underestimates specific humidity in the lowest level by ~4.9% (MERRA by 5.5%). Note that the error in reanalysis at BBM is relatively small because the Fennec radiosondes data were assimilated. In the subsequent absence of such observational data we expect reanalysis errors to be greater given the known problems of reanalysis model representation of meso-scale convective processes in the region (Garcia-Carreras et al., 2013; Roberts et al., 2015; Todd et al., 2013).

10    Such mesoscale convective 'cold pool' outflows (known locally as 'haboobs') are known to make a significant contribution to moisture advection as well as being the dominant dust emission process (Marsham et al., 2013b; Trzeciak et al., 2017). Red arrows in Figure 2a denote major haboob events.

### 3.1.2 Surface albedo

We calculate surface albedo from surface observations of shortwave flux at BBM for the days when good measurement is available (see Figure 3). During the days where measurements were not good, we use the diurnal average surface albedo of all other days. The mean surface albedo at BBM is 0.36 and shows strong diurnal cycle, varying with solar zenith angle giving maximum surface shortwave reflection during the morning and evening

20    hours, i.e. when the sun is at high solar zenith angles. This has an impact on the diurnal cycle of dust radiative effect (Ansell et al., 2014; Banks et al., 2014; Osipov et al., 2015).

### 3.1.3. Dust AOD

No observations of the vertical profile of dust loading at BBM are available from the Fennec instrumentation. Since the model requires the vertical distribution of mass mixing ratio of dust as an input, we use the long term mean extinction coefficient profiles for dust aerosol derived from the Cloud-Aerosol Lidar with Orthogonal Polarization (CALIOP) (Liu et al., 2009; Winker et al., 2009) satellite instrument. These are then scaled at each model time step to yield the observed column integrated AOD from the BBM AERONET

sunphotometer. We then use the mass extinction coefficient (in $m^2 Kg^{-1}$) to convert dust extinction coefficient (in $m^{-1}$) to dust mass mixing ratio (kg/kg) as required by the model (e.g. Greed et al., 2008). Mass extinction coefficient is calculated from Mie code (see Figure 7). Data from all individual CALIOP satellite orbits over the 2006-14 period were quality controlled and screened to retain dust aerosol only observations using the method described in Todd and Cavazos-Guerra (2016), which provides sampling for robust characterisation of aerosol

distribution in 3 dimensions (Ridley et al., 2012; Todd and Cavazos-Guerra, 2016; Winker et al., 2009). The long term mean extinction coefficient profile at BBM (Figure 4) shows a regular decrease of extinction through the aerosol layer which extends up to about 5 km at the top of the planetary boundary layer, which is also seen in Fennec airborne measurements from 2011 (Ryder et al., 2013a).

AOD data used to scale the mean extinction coefficient profiles are taken from retrievals form the

AErosol RObotic NETwork (AERONET) (Holben et al., 1998) instrument at BBM, using Level-2 data, which is





cloud screened and quality assured. We compared AERONET AOD with estimates of AOD from the SEVIRI instrument on Meteosat 9 satellite (derived from the 550nm channel using the algorithm of Banks and Brindley (2013)) over the June 2011 study period (Figure 5). The more frequent dust events during the latter half of the month (heat low phase) compared to the earlier heat maritime phase is apparent, with dust events frequently

associated with high water vapour indicative of convective cold pool 'haboob' dust events (see Figure 2a). The estimates of mean AOD agree to within 20% and there is a strong correlation between the two estimates of 0.7, despite some apparent dust events apparent in SEVIRI but not AERONET e.g. 13th and 29th June. This is mainly due to AERONET masking dust as cloud particularly in cases when dust and cloud coexist.

Nigh time dust emission is common during summer in the SHL region, although we expect dust

shortwave daytime radiative effect to be dominant (Banks et al., 2014). Estimation of AOD at night is problematic for most passive instruments and we use the following method: estimate AOD from observations of scattering from the nephelometer instrument deployed near the surface at BBM (Rocha-Lima et al., submitted), based on the regression of scattering to column integrated AOD during coincident daytime observations. The nephelometer-based estimates of AOD will account for night time emission of dust due to Haboobs (Marsham et al., 2013) but

since haboobs tend to occupy a shallow layer, than the better mixed daytime dust, this will tend to overestimate AODs estimated at night.

**3.2 RT configuration mode experiments towards 'optimal' configuration**

For some quantities, we do not have direct observations so we use alternative data from various sources. In the 'configuration mode' we test the uncertainty of the modelled radiative fluxes to uncertainties in these model inputs using the experiments summarised in Table 1. Then comparison of TOA fluxes with satellite observation allows us to arrive at what we consider to be an 'optimal' model configuration for the subsequent model

'experiment mode' analysis.

**3.2.1 Surface condition data**

Fennec does not provide observations of all the necessary information for the RT model and thus we

look for alternative best approximates from ERA-I and MERRA data of the following quantities:
(i) Surface skin temperature. Since there are no complete observations of skin temperature we use reanalysis products as inputs to the RT code and we use both these data to further investigate sensitivity of flux to uncertainty in skin temperature. Figure 6 displays the time series of surface skin temperature from ERA-I, MERRA, and CERES footprint data. Root mean square error (RMSE) of the reanalysis products with respect to CERES-

footprint data are high (4.5 K and 5.5 K for MERRA and ERA-I, respectively). Despite the higher RMSE of ERA-I skin temperature compared with RMSE of MERRA, the RMSE of ERA-I 2 m air temperature (Figure 6) with respect to flux tower measurement is 3.1 K (3.7 K, MERRA). The relatively bigger RMSE in skin temperature could be due to the uncertainty in CERES measurements.
(ii) Surface emissivity. We test the sensitivity of radiative fluxes to uncertainty in estimates of surface emissivity

using CERES measurements (mean=0.89) and MERRA outputs (mean =0.94).





(iii) Surface albedo. We noted that in contrast to observations the reanalysis products have weak representation of the diurnal cycle in surface albedo (Figure 3). Although we use observed surface albedo throughout all our experiment model RT runs, we also test the sensitivity of TOA shortwave flux to reanalysis surface albedo errors.

5   **3.2.2 Dust size distribution**

Dust radiative effect is known to be influenced by size distribution (Otto et al., 2009; Ryder et al., 2013a, b), which remains uncertain over the Sahara. We test the RT model sensitivity to two different and highly contrasting dust size distributions: (i) derived using AERONET sun photometer inversions from Cape Verde , representative of transported (Dubovik et al., 2002), referred to as Dubovik hereafter and (ii) measured directly from aircraft observations during the Fennec campaign (Ryder et al., 2013b) referred to as Fennec-Ryder hereafter, which include a pronounced coarse-mode with effective diameter in the range between 2.3 and 19.4 μm, contrasting with the much finer size distribution of Dubovik from AERONET. In both cases the dust size distributions and the same refractive index are used as inputs to Mie code in the RT pre-processor from which the optical properties of dust are calculated, specifically the single scattering albedo ($\omega$ or SSA), mass extinction coefficient (known as MEC or $K_{ext}$ units $m^2$ $Kg^{-1}$), and asymmetry parameter (g), for the relevant spectral bands applied in the RT model. Figure 7 displays the wavelength dependence of optical properties for both Dubovik and Fennec-Ryder dust size distributions. The continuous lines are the spectrally resolved optical properties and the horizontal lines are the band-averaged data which are used in the RT code. SSA values in the band covering the spectral range 0.32 to 0.69 μm are 0.82 and 0.91 for Fennec-Ryder and Dubovik respectively. The coarser particles in Fennec-Ryder result in a lower SSA – i.e. more absorbing dust. Note that in the model since the AOD is fixed based on the observed AOD, the vertical profile of dust mass mixing ratio is adjusted so that when combined with the MEC shown in Figure 7, the AOD in the spectral range 0.32 to 0.69 μm is correct. Therefore the differences in MEC between the two datasets shown in Figure 7 are cannot result in differences within the RT model. However, differences in SSA and g are able to exert different impacts on the radiative fluxes within the RT model, as described in section 4.1

**3.2.3 Cloud properties**

Acquiring observations of the vertical structure of clouds of sufficient quality for radiative transfer calculations is always challenging. Here we use the ERA-I and MERRA outputs of cloud fraction, liquid and ice water mixing ratios. Cloud is treated to have maximum overlap in a column where ice and water are mixed homogeneously. During the Fennec period, cloud was characterised by shallow cumulus or altocumulus near the top of the PBL and occasional deep convection. It is likely that the relatively coarse vertical and horizontal resolution of both reanalysis models will have considerable bias and we recognise that this is likely to underestimate the true cloud-related uncertainty, and for example, M16 suggest that ERA-I underestimate cloud fraction by a factor of 2.5.

**3.3 RT model 'experiment mode' design**

40





Using the suitably configured RT model (from Section 4.1) we addressed the research questions, specifically to quantify the combined and separate DRE of water vapour and dust, we undertook a number of experiments summarized in Table 2, i.e. the 'experiment mode'. For all the experiments RT calculations are made for each day using the atmospheric profiles at hourly time steps over the diurnal cycle, and the mean flux and

heating rates are derived by averaging outputs at each time step. All input data are linearly interpolated to a one-hour temporal resolution.

For the experiments with ('w') and without ('n') dust ('D') we simulate the 8th-30th June 2011 period. For the sensitivity ('sen') experiments, we simulate linearly increased levels of dust AOD and water vapour. We use runs both with cloud ('C') and with no cloud ('nC'). For dust sensitivity experiment ('senDnC'), AOD is increased

linearly over the range 0 (dust free) to 3 (extremely dusty), while keeping the mean value of water vapour constant. For water vapour sensitivity experiment ('senWVwDnC') the mean diurnal profile of water vapour is used but is scaled so that the column integrated water vapour increases from 10 to 40 kg m-2 and the mean AOD is used in each case.

The DRE for dust is derived by (i) subtracting TOA and surface fluxes of experiment wDnC from nDnC

(ii) linear regression of the flux dependence on the range of dust AOD from the dust sensitivity experiments (senDnC), in which a single diurnal cycle is simulated. The results are presented in Section 4.2. The impact of water vapour is determined by (i) composites of dry versus humid days from the nDnC experiment (ii) linear regression of the flux dependence on the range of water vapour from the water vapour sensitivity experiments (senWVwDnC).

## 4. Results and discussion

### 4.1. RT model optimum configuration and validation

Prior to testing the main research questions related to the relative roles of dust and water vapour in radiative heating (Section 4.2), the RT model was configured based on the 'configuration mode' sensitivity analyses (described in Section 3.1, Table 1) and comparison with observed TOA fluxes from the CERES-EBFA monthly mean product (clear sky in the case of all sensitivity analysis except the cloud sensitivity which we compare to CERES-EBFA all sky).

Sensitivity of RT simulated fluxes to uncertainty in the surface skin temperature and emissivity is low compared to the sensitivity to other factors (Table 1) with variations of ~2 W m-2 at TOA and 5-6 W m-2 at surface. Based on bias with respect to CERES-EBFA observations we use ERA-I skin temperature and MERRA emissivity products for the 'optimal' configuration.

Sensitivity to the two contrasting dust size distributions is pronounced. As expected results using Fennec-

Ryder dust show much stronger absorption in the shortwave compared with the Dubovik dust distribution, and the resulting TOA net shortwave flux is higher by 25 W m-2 in the former. These shortwave fluxes using Fennec-Ryder are not consistent with the GERB/CERES satellite observations (nor with previous estimates of shortwave DRE derived from satellite e.g. Yang et al. (2009); Ansell et al. (2014)) and we use Dubovik optical properties in the optimum configuration. Recent work suggests that the dust optical properties at BBM in June 2011 were

significantly less absorbing than both those measured by the aircraft further west during Fennec, and the Dubovik





representation (less absorbing, smaller sized) with SSA values of 0.99 (Rocha-Lima et al., submitted). Therefore, Dubovik optical properties represent intermediate values in terms of SW absorption.

TOA fluxes are not strongly sensitive to the choice of cloud properties with TOA net flux variations of ~4 W m$^{-2}$. On the basis of bias with respect to observations we select the ERA-I cloud properties. It is interesting to note that TOA radiative fluxes are quite sensitive to the errors in surface albedo from reanalysis with differences up to 16 W m$^{-2}$ compared to the optimum configuration, which used observed surface albedo. This suggests that it is important to have good observational data, which captures the strong diurnal cycle of surface albedo to achieve accurate radiative fluxes.

The RT model with the above choices of input data is considered to be the 'optimum' configuration, and we validate TOA and surface fluxes with respect to satellite and surface observations, respectively (Tables 3 and 4) for the most 'realistic' experiment wDwC. The sign convention used here and in the remainder of the paper is that downward flux is considered as positive while upward radiation is negative.

The simulated TOA net shortwave flux is 321 W m$^{-2}$, compared with 314 W m$^{-2}$ in GERB. It is -290 W m$^{-2}$ for net longwave, with -276 W m$^{-2}$ in GERB, giving 31 W m$^{-2}$ for net radiation, compared with 38 W m$^{-2}$ in GERB, i.e. there is more shortwave heating in the model, with more longwave cooling, giving less net TOA heating. These RT model shortwave/longwave/net (SW/LW/N) biases of 7/-14/-7 W m$^{-2}$ although larger than many of the sensitivity ranges for the input data uncertainties (Table 1) are within the estimated error of the GERB measurements (~15 W m$^{-2}$, Ansell et al., 2014). It is difficult to identify the most important sources of this bias although errors in the reanalysis skin temperature and ERA-I cloud representation included in the wDwC experiment are likely candidates. The DRE of cloud provides a useful comparison and could be considered to be an estimate of the upper limit of cloud-related biases. Cloud DRE (Table 5) is estimated from the difference in fluxes at the TOA between wDnC and wDwC to be -4/7/3 W m$^{-2}$ and from EBFA-CERES to be -15/16/1 W m$^{-2}$. These results of cloud DRE indicate that the optimum configuration flux biases are within the uncertainties of both observations and cloud effects.

At the surface there is a relatively wider disparity between simulated and observed flux (Table 4). The net shortwave simulated flux, 187 W m$^{-2}$, is 7 W m$^{-2}$ more than measured surface shortwave flux. Net longwave flux is -103 W m$^{-2}$ compared with that of measurement -78 W m$^{-2}$, the net effect being more cooling at the surface in the model than measurement by 25 W m$^{-2}$. We can again give comparison of cloud related biases between our result and CERES-EBFA product. Cloud SW/LW/N DRE at surface is estimated as -5/3/-2 W m$^{-2}$ from the wDwC-wDnC experiments and -19/11/-8 W m$^{-2}$ from EBFA-CERES, such that the shortwave bias at least could be explained by cloud but not all the longwave or net radiation bias. The remaining error could be attributable to measurement related errors and uncertainties to other variables such as surface emissivity, skin temperature, and surface albedo. Note also the difference in time averaging periods between the CERES-EBFA data covering whole of June 2011 and the RT experiments wDwC-wDnC covering for 8th-30th June could possibly contribute to part of the differences in the above figures.

RT model bias in the longwave is larger than that in the shortwave at both TOA and surface. The mean diurnal cycle of flux bias (Figure 8) shows that modelled outgoing longwave flux is overestimated at night time. Different factors could be attributed to this difference. Surface skin temperature used in this work is interpolated to 1 hr (Figure 6), which could lead to errors in the longwave flux. Satellite observations (see Marsham et al., (2013b)) show over both shallow cumulus cloud at the top of the PBL during late afternoon and occasional moist



convection preferentially at night, which may be missed in models given the poor representation of moist convection. This could also contribute to the difference between observed and calculated longwave flux associated with under-representation of cloud in the model.

The RT simulation wDwC captures well the day-to-day variability in radiative fluxes at TOA and surface (Figure 9) including the effect of the major synoptic and meso-scale dust/water vapour events e.g. the haboob event of 21st June. However, in the longwave there are significant RT model errors during the night time of 17th and 18th June, which are cases of high dust load following haboob events. Analysis of satellite imagery shows this anomalous high GERB longwave flux to be coincident with convective cloud development, presumably resulting from the moistening of the Saharan atmosphere, which the RT model, dependent on reanalysis cloud field, cannot

capture. This coincidence of dust and cloud is particularly challenging for both GERB cloud screening (which fails in this instance hence our use of all sky observations) and for the RT simulations themselves.

We can evaluate our model wDnC experiment results against clear-sky CERES footprint data in which RMSE are 17 W m$^{-2}$ and 12 W m$^{-2}$ for TOA shortwave and longwave fluxes, respectively. The equivalent figures for the model versus GERB (cloud screened using the CERES footprint cloud mask product) at the same times

are 22 W m$^{-2}$ and 12 W m$^{-2}$. These are comparable to and consistent with (i) the individual instrumental errors of CERES/GERB (ii) the inter-sensor uncertainties (CERES vs GERB RMSE = 22 W m$^{-2}$ and 6 W m$^{-2}$ for shortwave and longwave) (iii) previous similar studies (e.g. Osipov et al., 2015).

In summary, RT simulated flux errors of the 'optimum' configuration are comparable to observational uncertainties and those errors introduced by uncertainties in input fields. On this basis we suggest the RT

configuration is acceptable for further analysis on the direct radiative effect of dust and water vapour.

### 4.2. The radiative flux and heating effects of dust and water vapour

First, we consider the TOA and surface mean radiative budgets. In the absence of dust and cloud the

Saharan atmosphere during summer at BBM shows a positive radiation budget at the surface of 99 W m$^{-2}$ in which shortwave heating of 237 W m$^{-2}$ is offset by longwave cooling of -138 W m$^{-2}$ (Table 4). At TOA the shortwave flux of 328 W m$^{-2}$ is not quite offset by longwave losses of 313 W m$^{-2}$ (Table 3) leading to a net positive radiation balance of 15 W m$^{-2}$ making the SHL a weak net radiation sink. This strong (weak) radiation surplus at surface (TOA) leads to the atmosphere having a net cooling of 83 W m$^{-2}$ (i.e. radiative divergence), presumably

maintained by the transfer of sensible heat from surface into the atmosphere through turbulent heat transfer (Alamirew et al., submitted).

Both dust and water vapour are known to play an important role in controlling the radiative budget and heating rate of surface and the atmosphere over Sahara. Variability in these two active radiative components is strongly correlated due to the physical processes that govern transport of water vapour and dust emission into the

SHL region (Marsham et al., 2013b; M16) such that it is challenging to quantify their separate radiative effects from observations alone. Our RT simulations below address this research gap.

### 4.2.1 Dust





Here we determine the DRE of dust using two set of experiments described in Table 2. First we compare the simulations of radiative fluxes and heating during June 2011 between the wDnC and nDnC experiments (Figures 10, 11, 12, and 13 and Tables 3 and 4). Secondly, we derived the sensitivity of radiative fluxes and heating rates to a wide range of dust AOD using the sensitivity experiments (Figure 12). We then compare our

estimates of dust DRE to those from previous studies.

The mean SW/LW/N DRE of dust at TOA for June 2011 estimated from wDnC minus nDnC is -3/16/13 W m$^{-2}$ confirming the net warming effect of dust over the Sahara. This warming comes primarily in the longwave with a peak at ~24 W m$^{-2}$ close to midday (Figure 10a). The net shortwave DRE is small, consistent with other estimates (Huang et al., 2014; Yang et al., 2009). However, with a pronounced diurnal structure driven by a semi-

diurnal cycle in the shortwave with a cooling effect of up to -29 W m$^{-2}$ after dawn until 10:00 and after ~16:00 until sunset, and a warming effect of up to ~22 W m$^{-2}$ around midday (Figure 10a). The diurnal cycle of dust DRE is not strongly dependent on the amount of dust loading in the atmosphere but controlled by solar zenith angle and surface albedo (Ansell et al., 2014; Banks et al., 2014). The phase function also exerts a control on the diurnal cycle of the DRE as its value increases the backscatter fraction of SW radiation at large solar zenith angles. For

comparison, the equivalent TOA SW/LW/N DRE of dust for MERRA reanalysis are 10/7/17 W m$^{-2}$ suggesting that although MERRA has a good estimate of net DRE but the apparent shortwave warming effect is not in agreement with observations and the longwave warming is underestimated.

At the surface the SW/LW/N DRE of dust is estimated to be -45/32/-13 W m$^{-2}$ for SW/LW/N (Table 5). The net cooling is driven by the shortwave which peaks at ~-108 W m$^{-2}$ around noon (Figure 10b) partly

compensated by a longwave heating effect of 32 W m$^{-2}$. The MERRA reanalysis DRE at surface is -30/20/-12 W m$^{-2}$ again showing a good estimate of net effects but underestimating the shortwave and longwave components. The time series of shortwave DRE of dust (Figure 11a) at TOA further confirms the diurnal cycle discussed above: a midday warming and early morning and late afternoon cooling. The impact of big dust events (e.g. June 17[th] and 21[st]) can be clearly seen on the time series of longwave DRE of dust (Figure 11b).

The results of sensitivity experiments 'senDnC' are shown in Figure 12 and the DRE per unit AOD and per unit standard deviation in AOD is presented in Table 6, assuming a linear relationship between flux and AOD. We find the net TOA shortwave flux to be only weakly sensitive to dust AOD (Figure 12 (d) at -2 W m$^{-2}$ per AOD. This is due to the competing dust effect of increasing surface albedo which decreases net TOA shortwave and absorption by dust which increases TOA net shortwave by reducing the upwelling shortwave radiation. Our

estimates of shortwave dust DRE is less than half the sensitivity reported at BBM by M16, but consistent with the Sahara-wide estimates from satellite of Yang et al., (2009) and those of Ansell et al., (2014).

Dust in the atmosphere acts to reduce the outgoing longwave flux by 10 W m$^{-2}$ per unit increase in AOD (Figure 12a), warming the surface, consistent with the observations at BBM of M16 (11 W m$^{-2}$ per AOD increase) and within the Sahara-wide range of Yang et al., (2009). At the surface dust has opposing effect in shortwave and

longwave, with shortwave having stronger cooling effect: for every unit increase in AOD there is shortwave reduction (Figure 12e, Table 6) of 34W m$^{-2}$ compared to increase in longwave (Figure 12b) with 20 W m$^{-2}$ the net effect (Figure 12h) being cooling of -14 W m$^{-2}$ per AOD increase.

Dust drives radiative convergence in the atmosphere i.e. the difference in TOA minus surface flux, which acts to warm the atmosphere. This occurs through greater shortwave absorption, at a rate of 32 W m$^{-2}$ per AOD

(Figure 12f) offset partially by longwave cooling the atmosphere at -10 W m$^{-2}$ per unit AOD increase, leading to





a net warming effect of 22 W m⁻² per unit change in AOD. Overall, the RT estimates of TOA and Surface DRE in the shortwave and longwave and the atmospheric radiative convergence are within a few W m⁻² of those of M16 derived from observations.

We convert the radiative fluxes to actual heating rates (Figure 13a). In the absence of dust (nDnC experiment) the PBL is heated in the shortwave mainly from absorption by $O_2$ and water vapour peaking up to ~1.3 K day⁻¹ at 450 hPa (the water vapour effect is shown in Figure 15). Strong longwave cooling throughout the troposphere (up to ~-3 K day⁻¹ at ~450 hPa) due to emission from water vapour and other greenhouse gases exceeds this shortwave heating, leading to tropospheric radiative cooling of ~-0.6 K day⁻¹ throughout the PBL. This is consistent with the radiative heating estimate of Alamirew et al. (submitted) derived as a residual of the heat budget. In the lowest near surface layer (below 925 hPa) there is less longwave cooling due to strong radiative flux from the hot desert surfaces in the SHL. Dust acts to modify the vertical structure of this radiative heating/cooling considerably. Absorption of shortwave radiation leads to a strong warming effect in the shortwave (especially in the dusty PBL up to ~0.75 K day⁻¹ below ~700 hPa, where dust loadings are the highest), offset only partially by enhanced longwave cooling (up to ~-0.25 K day⁻¹) resulting in a net warming of the atmosphere by up to ~0.5 K day⁻¹ at ~700hPa, such that the dusty troposphere above ~600hPa has near zero cooling. For comparison we consider the MERRA reanalysis product mean heating rate (Figure 13b), which includes both cloud and climatological dust, is in close agreement with those of the wDwC experiment. However, MERRA does not capture the day-to-day variability in shortwave heating from dust and will not therefore be able to simulate the responses of the SHL atmosphere to variability at these timescales. Further weather/climate model simulations are required to determine the effect this has on the regional circulation and the behaviour of the SHL.

Day-to-day variability in the dominant shortwave net heating rate (Figure 14) is pronounced and shows the impact of the synoptic/meso-scale dust events on the SHL atmosphere. During large dust events (e.g. June 17ᵗʰ and 21ˢᵗ) there is strong shortwave heating up to 6 K day⁻¹ around midday hours. This will be coincident with reduced surface net radiation and sensible heat flux. Together these processes will act to reduce the vertical temperature gradient, stabilise the atmosphere, reduce dry convection and reduce the depth of the PBL.

### 4.2.2 water vapour

To estimate the heating rate profiles due to water vapour, we selected from the simulation nDnC the three driest days (June 11, 12, and 16) with mean column integrated water vapour of 20.2 Kg m⁻² and three most humid days (June 18, 25, and 30) with mean column integrated water vapour of 34.7 Kg m⁻². The mean heating rate profiles for the two contrasting atmospheric conditions is shown in Figure 15. High humidity drives an increase in the shortwave heating rate up to 0.5 K day⁻¹ peaking near the surface. This atmospheric warming is counteracted by a slightly bigger cooling in the longwave. Thus there is a net cooling up to -0.25 K day⁻¹ in atmospheric and strong heating up to 2.5 K day⁻¹ near the surface as a result of increase in moisture. The atmospheric cooling in the longwave causes surface warming, which is suggested to be linked with the intensification of the Saharan heat low region (Evan et al., 2015b). The reversed heating rate profiles in the layer between 500 hPa and 400 hPa is because of the mean moisture profile in this layer is larger during the dry days and the vice versa (figure 2).

Results from the water vapour sensitivity experiments 'senWVwDnC' are presented in Figure 16 and the linear dependence on fluxes per unit water vapour in Table 6. The outgoing longwave radiation (Figure 16a)



decreases with increasing of water vapour at a rate 1 W kg⁻¹ which is associated with the greenhouse effect of water vapour. This is about a third of that derived by M16 (3 W kg⁻¹). Their result includes the effect of water vapour and associated dust and cloud while our result considers sensitivity of radiative flux to changes in water vapour only. The sensitivity of TOA shortwave flux due to water vapour (Figure 16d) is 0.3 W Kg⁻¹ which warms

the atmosphere and to the contrary cools the surface due to the reduction of the shortwave reaching the earth. M16 showed that a reduction in the TOA shortwave radiation with increasing of water vapour, of -0.98 W Kg⁻¹ which is contrary to what we find in our results. But this could be related with to the impact of cloud on the shortwave radiation which will reduce the TOA net shortwave radiation. The net flux at TOA increases by to 1.4 W m⁻² for a unit change in CIWV resulting in a net warming of the TOA.

The net flux reaching the surface (Figure 16h) is increased at a rate 1.1 W Kg⁻¹ which is the counteracting effect of a dominant increase in longwave radiation re-emitted downwards from the atmosphere (1.5 W Kg⁻¹) and a smaller reduction in downwelling solar radiation (-0.4 W Kg⁻¹). The warming effect of water vapour at both the surface and the TOA means that net atmospheric convergence changes relatively little with water vapour (Figure 16i) at 0.1 W Kg⁻¹ which is a result of -0.6 W Kg⁻¹ in the longwave (Figure 16c) and 0.7 W Kg⁻¹ in the shortwave

(Figure 16f). In comparison to the observational analysis of M16 we see some important differences, notably we see a greater surface net warming effect of water vapour and as a result negligible, not positive atmospheric radiation convergence. Nevertheless our estimate of the sensitivity of surface longwave radiation to changes in CIWV of 1.1 W Kg⁻¹ is at the lower end of the range (1.0-3.6 W Kg⁻¹) estimated by Evan et al., (2015b), from observations and RT simulations, suggesting the role of water vapour in driving longer term interannual to decadal

heating of the SHL may not be as pronounced as previously suggested.

### 4.2.3. The relative effects of dust versus water vapour

From the sensitivity experiments, we can quantify the DRE of dust and water vapour at TOA and surface

25 per unit change in AOD dust and kg water vapour respectively (Table 6). By scaling this to observed standard deviation in each variable observed during the Fennec observation period we provide estimates of the relative importance of dust and water vapour to the day-to-day variability in the radiation budget over the SHL.

The resulting normalised dust SW/LW/net DRE per AOD at TOA and surface is -1/8/7 W m⁻² and -27/16/-11W m⁻² respectively, where these figures provide a useful way of presenting the variability of dust and

30 water vapour on their radiative effects. The equivalent values for water vapour are 2/6/8 W m⁻² and -2/8/6 W m⁻². As such, the radiative effects of dust and water vapour at TOA are of similar magnitude with net warming of ~7 W m⁻² per unit variability. Dust and water vapour exert similar control on the total heating of the Earth-atmosphere. This contrasts with M16 who report water effects (from vapour and cloud) as ~3 times more powerful than dust.

At the surface radiative flux is controlled much more strongly by dust than water vapour and with

35 opposite sign: net cooling of -11 W m⁻² and warming of 6 W m⁻² per unit variability respectively. M16 find near zero warming from water (vapour and cloud). In our study the net effect of TOA versus surface is strong atmospheric warming of 18 W m⁻² per unit variability from dust and negligible warming (1 W m⁻² per unit variability) from water vapour. In contrast, M16 find almost equal warming from dust and water vapour (of 11-12 W m⁻² per unit variability). Although this radtiative transfer based analysis of the role of water vapour does not





include the cloud that is implicitly included in M2016, we suggest that the co-variability of dust and water vapour hinders calculation of their independent effects in the observational analysis of M16.

In summary we find that dust and water vapour exert a similarly large control on TOA net radiation and therefore total column heating and by implication to the first order similar control on surface pressure in the SHL.

However, the vertical structure of radiative heating from dust is far more complex than that for water vapour. The schematic, Figure 17 illustrates the sensitivity of dust and water vapour at different pressure levels. Dust imposes a strong net cooling at the surface from the SW which declines to zero at ~700hPa, where SW cooling and LW warming balance, with net warming above this (Table 6). In contrast water vapour imposes a LW heating effect that varies relatively little from surface to TOA. As such dust is likely to have stronger impact on the structure

and processes of the SHL atmosphere than does water vapour.

**5. Summary and Conclusions**

The summertime Saharan Heat Low feature is of considerable importance to the wider regional climate

over West Africa but remains poorly understood. To the first order the SHL is created by strong sensible heat flux from the surface radiative surplus which heats the deep Saharan boundary layer, which is in near balance with advective cooling from the low level convergence circulation. However, radiative heating is modulated by water vapour and dust whose variations, at least at short timescales, are correlated. Here, we aim to quantify the independent radiative effects of dust and water vapour, and the vertical profile of atmospheric heating rates using

an RT model. The model is configured for the location at BBM, close to the heart of the SHL using inputs from Fennec field campaign for June 2011. First, sensitivity to uncertainty in RT model inputs fields is assessed. We find that dust size distribution is the most important source of uncertainty in this case, through its impact on single scattering albedo. The RT model when suitably configured has radiative flux biases at TOA that are within observational uncertainties and input uncertainties. The subsequent RT experiments show:

1. On average the SHL has a large positive radiative surplus at surface of 83 W m$^{-2}$, a small surplus at TOA of 31 W m$^{-2}$ with a net atmospheric radiative divergence of 52 W m$^{-2}$ presumably approximately balanced by the transfer of sensible heat.

2. The effect of dust is pronounced:

I.    During June 2011 dust had a positive DRE at TOA of 8 W m$^{-2}$ per unit AOD (7 W m$^{-2}$ per unit AOD

variability) almost entirely in the longwave, as the effects of shortwave absorption with respect to surface albedo largely balance, acting to warm earth-atmosphere systems as a whole, with magnitude consistent with previous studies (Banks et al., 2014; M16; Yang et al., 2009).

II.    Dust has a strong negative DRE at the surface of -14 W m$^{-2}$ per unit AOD (-11 W m$^{-2}$ per unit AOD variability) largely due to reduced shortwave flux from atmospheric absorption.

III.    The net effect of this negative surface DRE and positive TOA DRE is considerable atmospheric radiative convergence of 22 W m$^{-2}$ per unit AOD (18 per unit AOD variability) largely from shortwave absorption. This directly heats the PBL below ~500hPa by ~0.6 K day$^{-1}$.

IV.    Dust loading is variable and the heating effect of episodic synoptic and meso-scale dust events can be up to 6 K day$^{-1}$.

3. The effect of water vapour is weaker than dust at the surface and includes:





    I.     a positive radiative effect at TOA of 1.4 W m$^{-2}$ per unit column integrated water vapour (8 W m$^{-2}$ per unit water vapour variability) almost entirely a longwave greenhouse effect.

    II.     a weak positive radiative effect at the surface of 1.2 W m$^{-2}$ per unit column integrated water vapour (6 W m$^{-2}$ per unit water vapour variability) almost entirely from longwave radiation re-emitted downwards.

    III.     positive radiative effects at surface and TOA and thus a negligible impact on atmospheric radiative convergence.

A key finding here is that in contrast to previous analysis dust and water vapour are roughly equally important at the TOA, in controlling day-to-day variability in heating the earth-atmosphere system as a whole, (in contrast to M16 who identify water and associated cloud as the key driver), but that dust variability dominates variations in surface and atmospheric radiative heating. The biggest single net radiative effect in this study is the atmospheric radiative convergence from dust. The impact of dust may therefore be greater than previously believed. Recent studies have proposed a water vapour positive-feedback mechanism driving decadal variations in SHL intensity, implicated in the recent recovery of Sahelian rainfall (Evan et al., 2015b). Our results are consistent with this but strongly suggest that variability in dust loading should be considered in explaining variability and change in the SHL, reinforcing the need for high quality long term aerosol observations. Additionally dust size distributions, shape and chemical composition are spatially and temporally variable, and the vertical profile of dust will vary with meteorological conditions – thus introducing more variability and uncertainty than has been explored in this study. These variations potentially increase the controls of dust on the radiation budget even further than quantified here.

Therefore, water vapour events in themselves act to heat at the TOA and at the surface and presumably intensify the SHL. In contrast, dust events cool the surface but warm the lower troposphere as a whole, such that the net effect at the top of the Saharan residual layer (about 5km) is a warming which will intensify the SHL. Various climate model experiments also demonstrate this effect (Mulcahy et al., 2014). We can then consider the effects of variability in SHL associated with monsoon surges and haboobs in which dust and water vapour increases are often coincident. Through radiative processes such events act to (i) warm the whole troposphere, almost equally through dust and water vapour longwave effect (ii) strongly cool the surface from dust shortwave effect, and more weakly warm the surface through water vapour longwave effects. Although these counteracting effects mean the net surface radiative flux reduction is actually quite small, the diurnal effects are pronounced with the dust shortwave apparent in daytime and the water vapour effect dominant at night, which will act to reduce the sensible heat flux into the atmosphere limiting the vertical development of the SHL PBL (iii) Substantial radiative heating from dust occurs in the PBL up to 6 K day$^{-1}$ through dust shortwave absorption. This will act to stabilise the PBL with implications for dry and moist convection, although Trzeciak et al. (2017) suggest that moistening may often counter this. Such events typically involve an additional advective cooling which can be substantial up to 2-5 K day$^{-1}$ for monsoon surges (Couvreux et al., 2010) but is restricted to the lowest layers ( ~1 km from surface).

Couvreux et al. (2010) suggested a negative feedback process within the SHL-monsoon systems that may govern preferred 3-5 day timescale of variability in the SHL and monsoon pulses. Strong net radiative heating at the surface intensifies the SHL, enhancing monsoon surges which then, through low level advective cooling, act to weaken the SHL, before solar heating restores the SHL. Our results add potentially important detail





regarding the radiative role of dust and water vapour that may modify this conceptual understanding. First, the net effect on surface radiation of dust and water vapour together is to further cool the surface and weaken the SHL, in addition to the advective cooling. Second, this weakening of the SHL is offset because the magnitude of dust radiative heating in the lowest layers is comparable to that of advective cooling so that net effect may be small or

even positive, but with the dust radiative heating extending throughout the entire PBL above, rather than just the lowest 1km or so. Third, the timescale of re-establishment of the SHL through surface heating and sensible heat flux may be influenced by the rate of dust deposition and export, which, depending on the synoptic context may be 1-2 days, though sometimes dust remains suspended in the SHL for days -weeks. The net effect of these, often competing, processes on the SHL will depend on the precise nature of water vapour, dust and temperature

advection during such monsoon surge events. As such, SHL variability will represent a complex interplay of factors rather than a feedback through a single mechanism. There is a clear need for much better spatially extensive and detailed observations of all these variables

We can therefore envisage an inherent tendency for pulsing in the SHL in which an intensifying SHL will lead towards monsoon surges, which act through near surface/low level radiative and advective cooling to

weaken the SHL and through dust-radiative heating to stabilise the PBL, until dust deposition and export allow re-warming of the surface to re-invigorate the SHL.

Given the radiative effects described above the dynamical effects of dust variability on the low level convergence and mid-level divergence circulations will be greater than those of water vapour and require further model experiments to resolve. Whilst reanalysis models represent well the average radiative and heating effect of

dust and water vapour they do not capture dust and water vapour variability well and the subsequent dynamical effects on the larger scale circulation.

The unique observations of the Fennec aircraft campaign suggested that fresh dust is much coarser than previously believed (Ryder et al., 2013b), with corresponding higher absorption, and this has significant impacts on the radiation budget (Kok et al., 2017). Our RT model simulations results suggest that such a dominant coarse

mode is not consistent with TOA radiative flux observations at BBM. However, if dust is coarser than we assume here then the radiative effects of dust would be even stronger. Further observations on dust size distribution and optical properties are a priority requirement. In addition, further work should consider in much greater detail the radaitive effects of cloud based on detailed observations rather than the rather coarse estimates from reanalysis used here.

Our results showing the complex interplay of dust and water vapour on surface and PBL radiative heating stress the need for improved modelling of these processes over the SHL region to improve predictions including those for the WAM across timescales (e.g. Evan et al., 2015). Most models currently struggle in regard to short term variability in water vapour (Birch et al., 2014; Garcia-Carreras et al., 2013; Marsham et al., 2013a; Roberts et al., 2015), clouds (Roehrig et al., 2013; Stein et al., 2015) and dust (Evan et al., 2014), with many dust errors

coming from moist convection ( Heinold et al., 2013; Marsham et al., 2011). Forecast models typically have mean biases of up to 2 kg m$^{-2}$ in column integrated water vapour (equivalent to change in 2.6 W m$^{-2}$ TOA net flux) and lack variability in dust, and so are expected to poorly represent these couplings. A focus on improved representation of advection of water vapour, clouds and convection in models should be a priority.



**Acknowledgements**

This research was funded through the Peter Carpenter African Climate scholarship, further supported by UK
NERC consortium grant NE/G017166/1. We would like to acknowledge data provided from NERC-funded
FENNEC project. Helen Brindley has given us constructive suggestions that were important to complete this
research. We thank Jamie Banks for providing processed GERB TOA radiation data and Adriana Rocha-Lima for
providing Nephelometer measured dust optical properties data. We would like to thank Azzendine Saci,
Abdelkader Ouladichir, Bouzianne Ouchene, Mohammed Salah-Ferroudj, Benyakoub Abderrahmane,
Mohammmed Limam, and Diali Sidali (ONM) and Richard Washington (University of Oxford) for their
contributions to setting up running the Fennec supersite, and indeed all at ONM Algeria for their patience and
hospitality during Fennec. We would like to thank the AERONET PHOTONS team for their assistance with the
Cimel Sun photometer. The authors also would like to thank Eleanor Highwood for facilitating access to
computational cluster at Metrology Department in University of Reading.

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

over Sahara. *Geophysical Research Letters,* 36.

**Figure Captions**

**Figure 1**. Climatological state of the Saharan heat Low region (mean JJAS from 1979-2013): SHL location, low
level circulation, and dust load. Shaded: the mean position of heat low region (occurrence frequency of 90% of
llat), arrows: mean 925 hPa wind, Blue Line: the mean position of the inter-tropical discontinuity from ERA-Int
reanalysis data and aerosol optical depth (AOD) from satellite MISR data (contour intervals are 0.4, 0.6, and 0.8
for grey, white, and cyan lines). The purple rectangle denotes location of the FENNEC Supersite 1 (SS1)

**Figure 2**. Vertical Profile Specific Humidity (a) FENNEC radiosonde measurements (b) ERA-INT and (c)
Difference between (a) and (b). Red arrows in (a) denote times of major haboob events

**Figure 3**. Diurnal Cycle of mean Surface Albedo at BBM

**Figure 4**. Caliop mean Extinction Coefficient profile at BBM 2006-13

**Figure 5**. AOD from AERONET and SEVIRI, and column integrated water vapour from FENNEC observation.
Gray shades show driest (11, 12, and 16), blue shades shows most humid days (18, 25, and 30)

Nephelometer measurement and green shade shows a major haboob event occurred on the 21[th] which resulted
in large dust emission

**Figure 6**, Surface skin temperature (SKT) and 2 m air temperature at BBM: Skin Temperature Black: ERAI, Red:
MERRA, and Green Star: CERES footprint, 2 m air temperature Gold:ERAI and Cyan: Flux Tower measurement.
The black and red stars denote ERAI and MERRA skin temperature at the time steps when there is CERES
observation.



**Figure 7**. Wavelength dependence of optical properties of dust particle for longwave (top three panel) and shortwave (bottom three panel). The continuous lines are the spectrally resolved optical properties and the horizontal lines are the band-averaged data that are used in the RT code.

**Figure 8**. Mean Diurnal Cycle of TOA Flux. Blue: SOCRATES wDwC and green: GERB

**Figure 9**. Time series of TOA (left column) and Surface (right column) shortwave (SW), longwave (LW), and net Radiative Flux at BBM. Black lines denote SOCRATES outputs, red line denote GERB measurements, green dots denote CERES measurements and red dots denote GERB measurements corresponding to CERES time steps.

**Figure 10**. Mean Diurnal direct radiative effect of dust averaged for June 08-30, 2016. TOA DRE of Dust (a) and Surface DRE of Dust (b) The bars show standard error over the diurnal cycle.

**Figure 11**. DRE due to Dust: time series of TOA shortwave (a), longwave (b), net (c) and surface shortwave (d) longwave (e) and net (f) wdnC-nDnC flux

**Figure 12** Radiative budget as a function of dust AOD. Top row (a, b, c): TOA longwave (a), shortwave (b), and net (c). Second row (d, e, f): similar to top row but for surface. Third Row: atmospheric radiative convergence of longwave (g), shortwave (g), and net (i)

**Figure 13**. Mean Radiative Heating Rate Profile for June (08-30, 2011) at BBM a: Results from nDnC (dashed lines) and wDnC (solid lines) using FENEC profile and b: MERRA Model output for all sky (solid lines) and clear sky (dashed lines) conditions. Blue, red, and green colours represent shortwave, longwave, and total heating rates respectively.

**Figure 14** Radiative Heating rates (K.Day$^{-1}$) of dust in the atmosphere (DUST represents wDnC runs and CLEAN represents nDnC runs)

**Figure 15** Atmospheric Heating rate profile for selected dry days, June 11, 12, and 16 (dashed lines) and moist days, June 18, 19, and 25 (solid lines)

**Figure 16** Same as Figure 12 except for column integrated water vapour.

**Figure 17** Sensitivity of Radiative Flux (W.m$^{-2}$) to changes in dust AOD and column integrated water vapour. The numbers at each pressure level are downward shortwave (blue), longwave (red), and net (green) flux. The grey shade represents dust and water vapour amount in the atmosphere





**Tables**

**Table 1**. Summary of model configuration mode sensitivity analysis

| Sensitivity input variable | Source of data for sensitivity run | Sensitivity results | 'Optimal configuration choice |
|---|---|---|---|
| **surface albedo** | Fennec measured quantity versus ERA-I | Difference of upto 16 W.m$^{-2}$ in TOA net SW flux | Surface Albedo calculated from surface flux measurements |
| **skin temperature** | ERA-I MERRA | Difference of 6 W.m$^{-2}$ in surface net LW flux | ERA-I skin temperature estimate |
| **Surface emissivity** | CERES MERRA | Differences of 2.3 W.m$^{-2}$ at TOA LW flux and 5 W.m$^{-2}$ at the surface. | MERRA reanalysis estimates |
| **Cloud fraction and mixing ratio** | ERA-I MERRA | Difference of 4 W.m$^{-2}$ both at TOA and surface net SW flux | ERA-I |
| **dust size distribution** | Dubovik FENNEC-Ryder | TOA SW dust DRE -2 w.m$^{-2}$ Using Dubovik and 23 w.m$^{-2}$ using Ryder-FENNEC | Dubovik |

**Table 2**. Description of the RT 'experiment mode'. Names of different experiments acronyms are defined as 'n'
5  = NO, 'w' = with, 'D' = Dust, 'C' = Cloud, 'WV' = water vapour, and 'sen' = sensitivity

| Name | Description | Water vapour | Aerosol | Cloud |
|---|---|---|---|---|
| nDnC | Dust free and Cloud free atmosphere | Observed 8$^{th}$-30$^{th}$ June 2011 | None | None |
| nDwC | Dust free but cloudy atmosphere | Observed 8th-30th June 2011 diurnal cycle | None | ERAI MERRA |
| wDnC | Cloud free but dusty atmosphere | Observed 8th-30th June 2011 diurnal cycle | AERONET AOD scaled with CALIOP Extinction | None |
| wDwC | Dusty and Cloudy Atmosphere | Observed 8th-30th June 2011 diurnal cycle | AERONET AOD scaled with CALIOP Extinction | ERAI MERRA |
| senDnC | Sensitivity to full range of possible AOD | Mean diurnal WV | Linear increase in AOD 0.0 to 3.0 | None |





| | | | Constant AOD each time step for a given run | |
|---|---|---|---|---|
| senWVwDnC | Sensitivity to full range of possible WV | Linear increase in TCWV from 10 to 40kg.m-2 at 2.5 kg.m$^{-2}$ interval with mean diurnal WV profile | Mean Diurnal AOD | None |

**Table 3**. Mean June 08-30, 2011 TOA Radiative flux at BBM (definition of acronyms are given in table 2). Values are in W.m$^{-2}$. The sign convention is that downward flux is considered as positive while upward flux is negative.

| | | nDnC | nDwC | wDnC | wDwC |
|---|---|---|---|---|---|
| TOA_SW | SOCRATES | 328 | 322 | 325 | **321** |
| | GERB | -- | -- | -- | **314** |
| | MERRA | 312 | 307 | 322 | **317** |
| | ERAI | -- | -- | 336 | **324** |
| TOA_LW | SOCRATES | -313 | -304 | -297 | **-290** |
| | GERB | -- | -- | -- | **-276** |
| | MERRA | -314 | – | -307 | **-296** |
| | ERAI | -- | -- | -309 | **-294** |
| TOA_NET | SOCRATES | 15 | 18 | 28 | **31** |
| | GERB | -- | – | -- | **38** |
| | MERRA | -2 | – | 15 | **20** |
| | ERAI | -- | -- | 27 | **29** |

**Table 4**. Same as Table 3 but for surface radiative flux and observation from fennec instrument

| | | nDnC | nDwC | wDnC | wDwC |
|---|---|---|---|---|---|
| SRF_SW | SOCRATES | 237 | 232 | 192 | **187** |
| | FENNEC_OBS | -- | -- | -- | **180** |
| | MERRA | 220 | 215 | 190 | **185** |
| | ERAI | -- | -- | 210 | **199** |





| | | | | | |
|---|---|---|---|---|---|
| SRF_LW | SOCRATES | -138 | -134 | -106 | **-103** |
| | FENNEC_OBS | -- | -- | -- | **-78** |
| | MERRA | -139 | -- | -119 | **-115** |
| | ERAI | -- | -- | -139 | **-132** |
| SRF_NET | SOCRATES | 99 | 98 | 86 | **84** |
| | FENNEC_OBS | -- | -- | -- | **103** |
| | MERRA | 82 | -- | 70 | **70** |
| | ERAI | -- | -- | 71 | **67** |

**Table 5**. TOA and Surface Direct Radiative Effect of Dust and Cloud

| | Dust DRE | | | Cloud DRE | | | | | |
|---|---|---|---|---|---|---|---|---|---|
| | SOCRATES | | | SOCRATES | | | EBFA-CERES | | |
| | SW | LW | NET | SW | LW | NET | SW | LW | NET |
| TOA | -3 | 16 | 13 | -4 | 7 | 3 | -15 | 16 | 1 |
| SURFACE | -45 | 32 | -13 | -5 | 3 | -2 | -19 | 11 | -8 |

**Table 6**. Sensitivity of Radiative Flux to Dust AOD and CIWV at selected altitudes.

5    SD*=Standard Deviation (0.8 for AOD and 5.5 g.kg$^{-1}$ for water vapour. Mean AOD = 1.2 and mean column integrated water vapour = 27.8 Kg.m$^{-2}$)

| Change in Flux | | SW | LW | NET |
|---|---|---|---|---|
| per unit AOD (W.m$^{-2}$) | TOA | -1.8 | 10.0 | 8.2 |
| | Surface | -33.8 | 19.8 | -14.0 |
| | Convergence | 32.1 | -9.7 | 22.4 |
| per unit CIWV (W.Kg$^{-1}$) | TOA | 0.3 | 1.1 | 1.4 |
| | Surface | -0.4 | 1.6 | 1.2 |
| | Convergence | 0.7 | -0.6 | 0.1 |
| per one AOD SD* (W.m$^{-2}$) | TOA | -1.4 | 8.0 | 6.6 |
| | 500hPa | -6.2 | 10.6 | 4.4 |
| | 700hPa | -14.8 | 11.6 | -3.2 |
| | Surface | -27.0 | 15.8 | -11.3 |
| | Convergence | 25.7 | -7.8 | 17.9 |
| | TOA | 1.7 | 5.8 | 7.5 |



| per one CIWV SD$^*$(W.Kg$^{-1}$) | 500hPa | -0.4 | 9.3 | 8.9 |
| | 700hPa | -1.6 | 9.4 | 7.9 |
| | Surface | -2.4 | 8.3 | 5.9 |
| | Convergence | 4.0 | -2.8 | 1.3 |

**Figures**

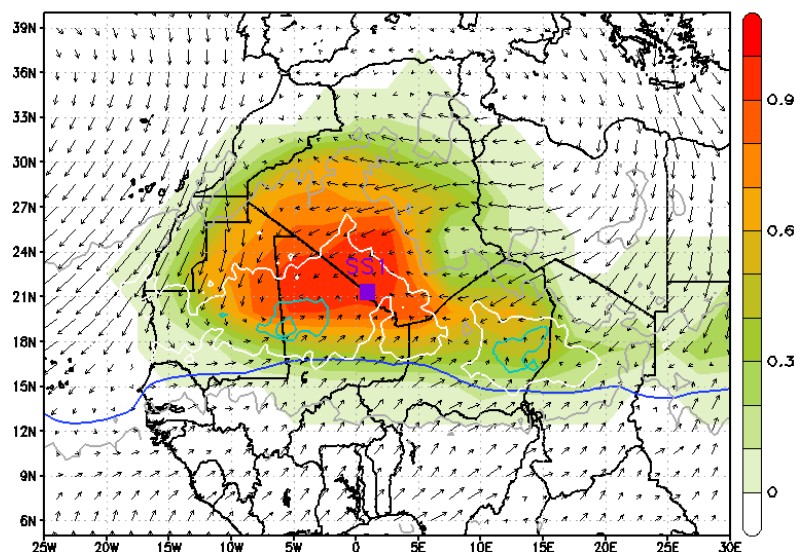

5     **Figure 1**



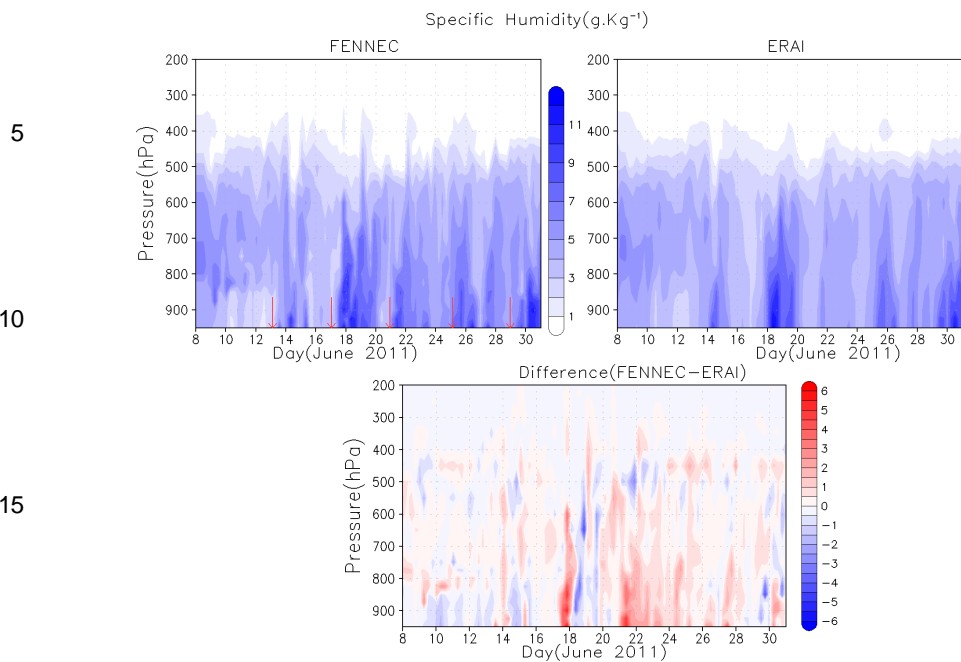

**Figure 2**

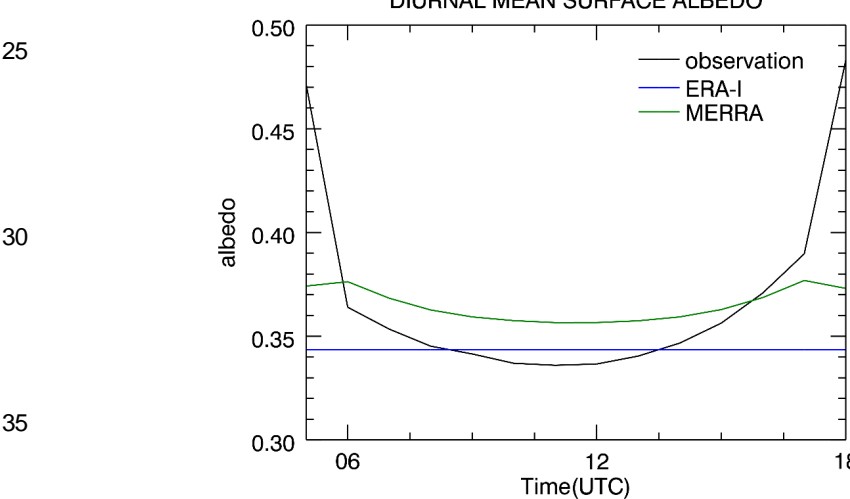

**Figure 3**

40





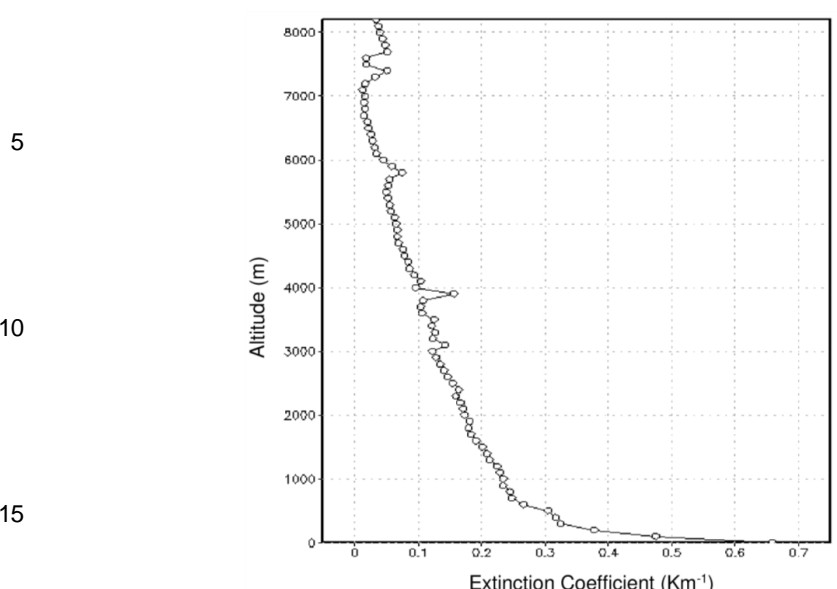

**Figure 4**

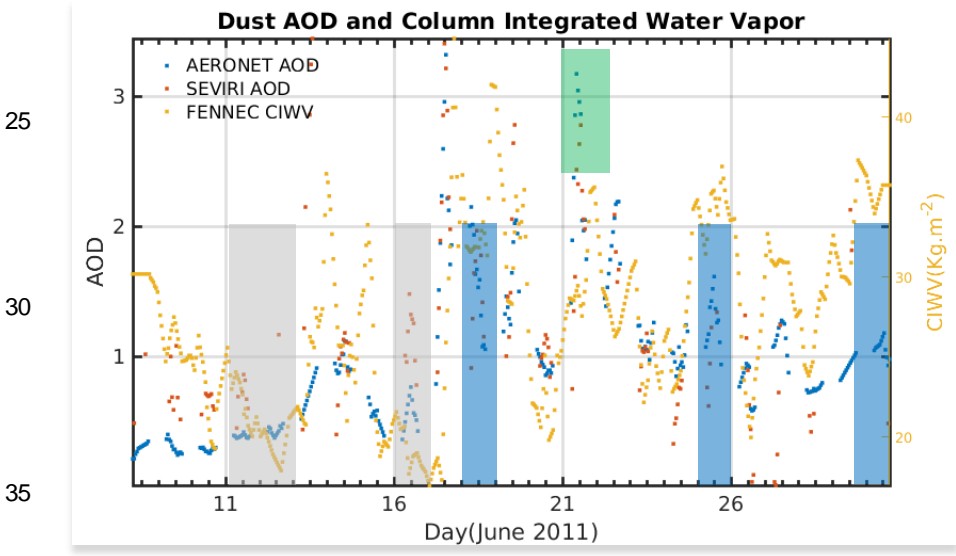

**Figure 5**

40



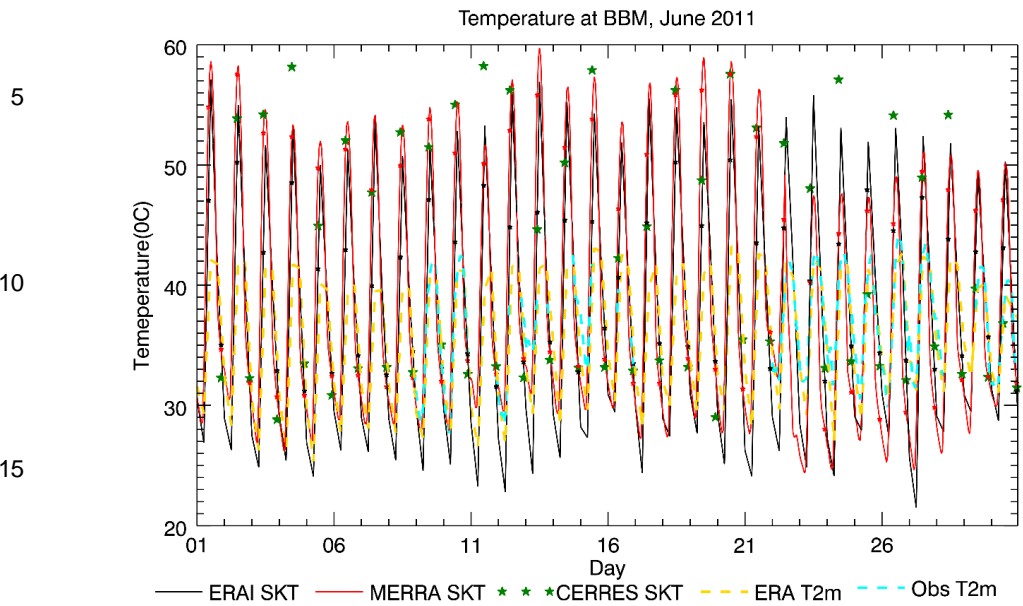

Figure 6

40







**Figure 7**



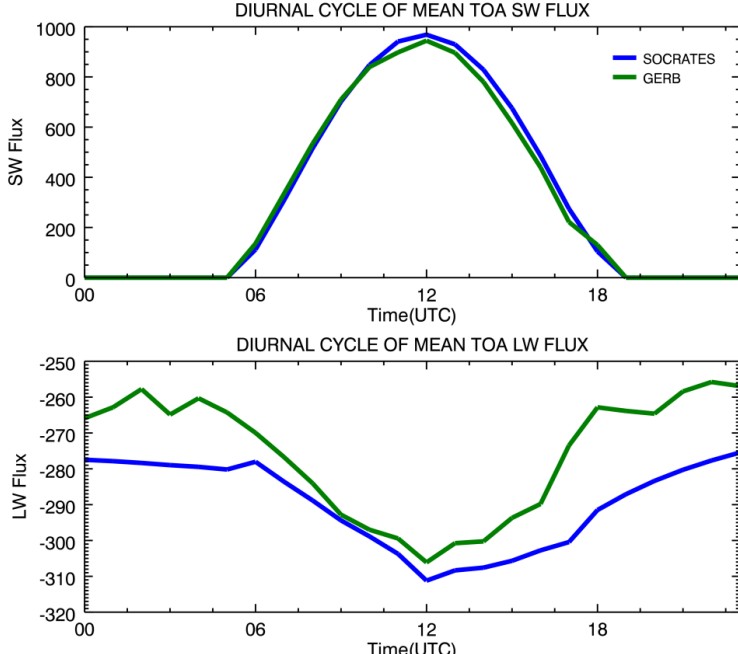

**Figure 8**

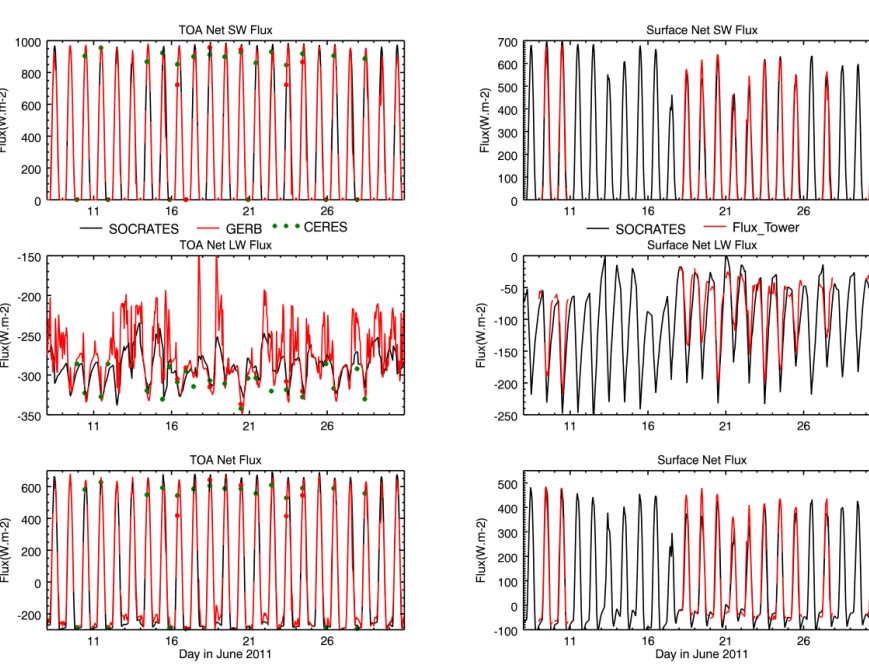

**Figure 9**





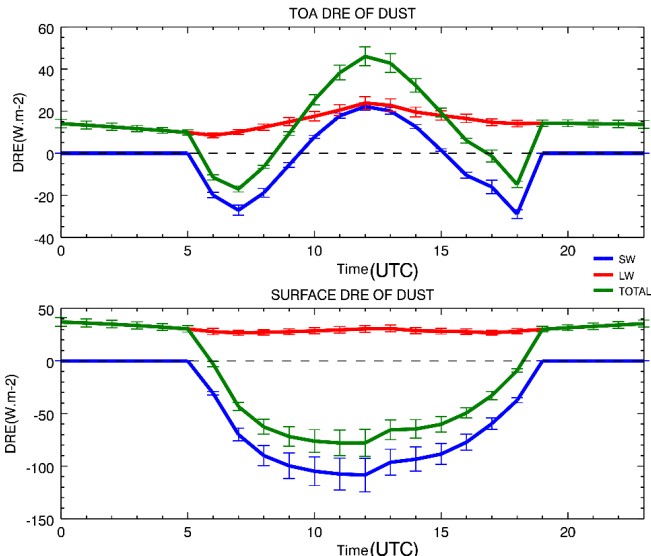

**Figure 10**

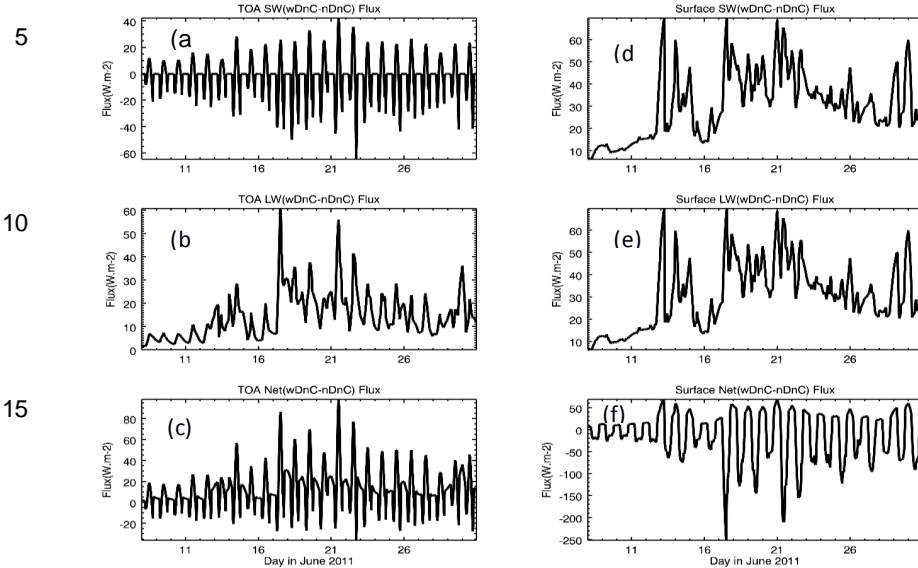

**Figure 11**



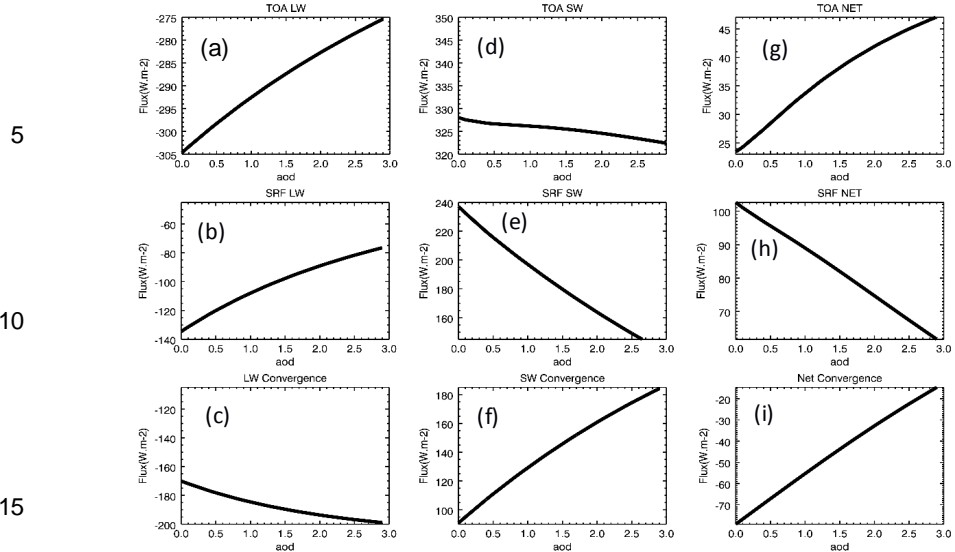

**Figure 12**

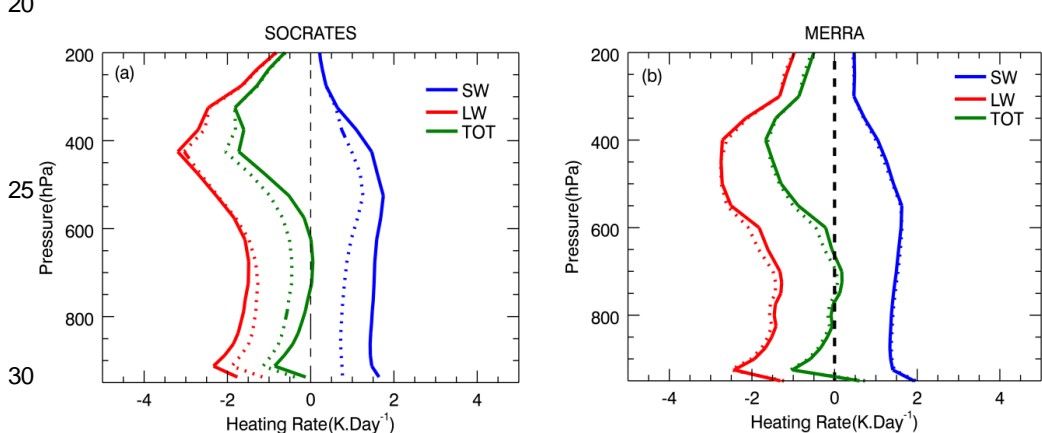

**Figure 13**





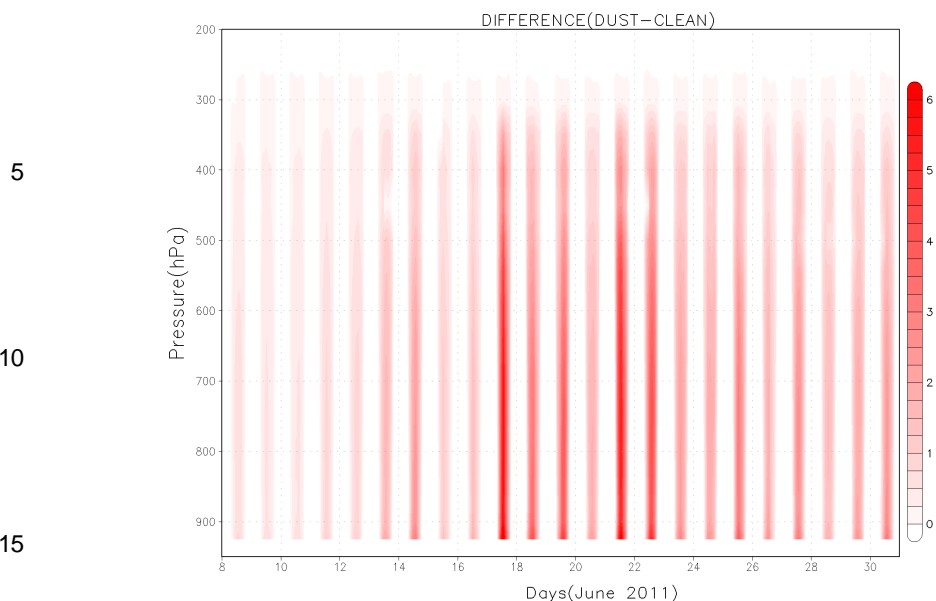

**Figure 14**

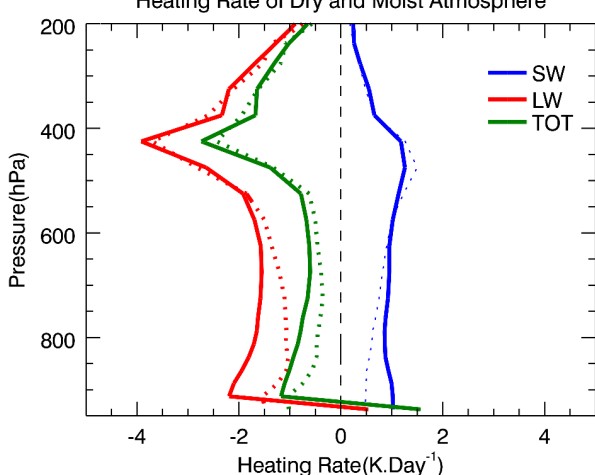

**Figure 15**





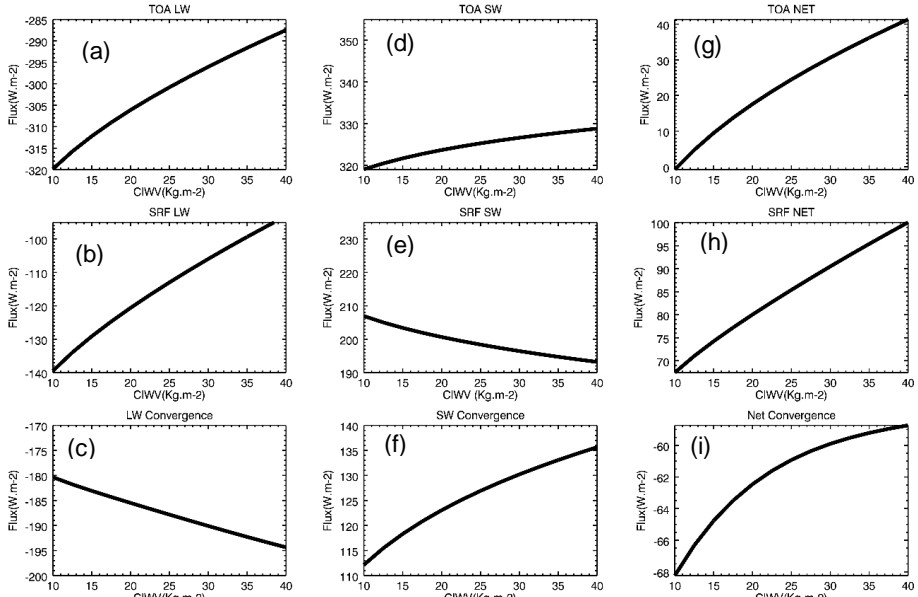

**Figure 16**

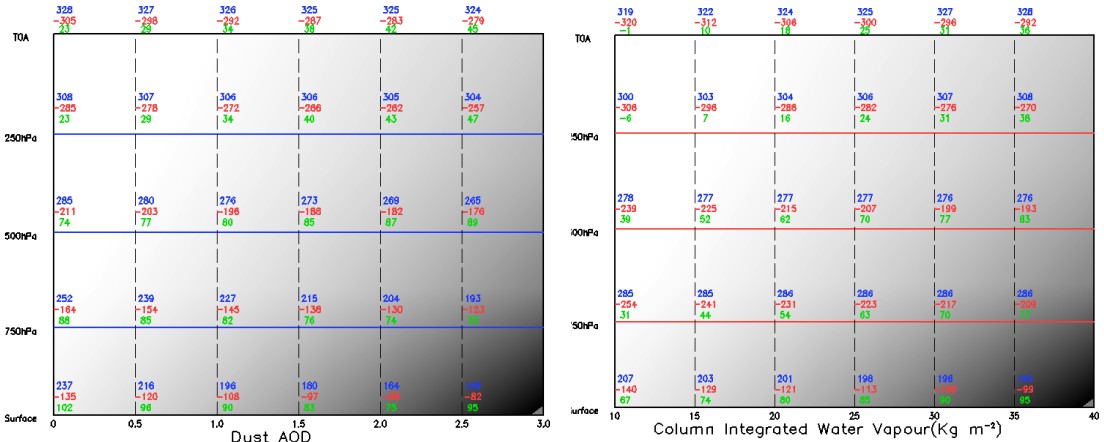

**Figure 17**