# Peer review of "The Early summertime Saharan heat low: Sensitivity of the radiation budget and atmospheric heating to water vapor and dust aerosol."

_Atmospheric Chemistry and Physics, 2017_

## Short Comment (SC1) · 29 May 2017

Review of the paper "The summertime Saharan heat low: sensitivity of the radiation budget and atmospheric heating to water vapor and dust aerosol" by N.K. Alamirew et al.

This study aims at assessing the impacts of dust aerosols and water vapor on the radiation budget during June 2011. After an introduction, the authors describe the radiative transfer model used and the observations for the validation. Except few minor points, these sections are clear and well written. I have more difficulties with the section 3, which is for technical and by construction not clear at all. Section 4 describes the main results of the simulations. Most of the time, these results are very descriptive and

it is hard to follow the authors. Finally the last section sums up and concludes this study. Here, some conclusions are too speculative considering the period of study and some approximations. After a careful reading, I consider this study relevant for the journal with innovative results that bring new insights. Nevertheless, I would recommend major revisions before to accept this paper. Please find the major and minor comments below.

Major comments: a- Section 3 is not clear. It is quite complicated to understand all the configurations and the conclusions drawn from these results on the choice of certain parameters. Finally the choices are not really justified and I am not sure it is necessary to provide all the information. I would recommend to simplify this section and to put some results in supplementary material. In this section I also found some parts not clear: p5 I5-12; it is quite weird to compare observations assimilated with model datasets. The authors do not explain the remaining errors. Is it due to the assimilation procedure?

b- Section 4 is too descriptive with too much information that are not necessarily significant or important to the conclusions of this study. This is particularly true p9 and 10. I strongly recommend to reduce this part to the most important results and to put the others results into an annex. The summary of the subsection 4.1 is too speculative. How the authors can conclude the simulated flux errors of the optimal configuration are comparable to the observational uncertainties? What does 'acceptable' mean?

c- Some conclusions are too speculative. The authors conclude about the impacts of the dust aerosols and water vapor on the SHL but, in that study, only June 2011 is used. The SHL is the most important from end of June to mid of September (when it is installed in its Saharan location). Even if the authors used only one month (June), they have to characterize this specific year to the climatology (in term of dust, humidity, large scale forcings). This point concerns the title ('summertime' is not appropriate), the conclusions (p15 l8-10), and the abstract. Also the discussion on the impacts on the SHL pulsations should be carefully discussed since the authors do not analyze the contribution of the large scale temperature advections and they never show the
real position of the SHL in June 2011 (in June, the SHL is migrating to the north with a large spatial variability). Finally at climatological scale, the authors should pay attention to the climatological evolution of the dust that tends to reduce (p15 I16).

d- Some figures are not readable.

Minor comments

P2 I11 the authors should mention this reference: Lavaysse, C., Flamant, C., Evan, A. et al. Clim Dyn (2016) 47: 3479. doi:10.1007/s00382-015-2847-z

P6 I4; the two phases mentioned are not so clear.

P6 I19: title of subsection 3.2 not clear, please rephrase

P6 l24: optimal to what?

P6 I37-38; how do the authors conclude the Ceres measurements are uncertain and that explain the large RMSE? The term RMSE refers to a reference (usually observations) that are considered as the correct value. Here, I do not understand what is the reference and how they can conclude that. Please clarify. Also the term RMSD (difference) should be more appropriate.

P6 I39-40: the authors provide some results without explanations, what are these results (mean =  $\dots$ ) and please clarify the conclusions/interest of this point?

P7 subsection 3.2.2 I recommend to put the first part of the paragraph in the introduction section and the result in supplementary material.

P8 I1: Section 4.1 is correct?

P8 I11: Is it necessary to use this acronym?

P8 I27: Section 3.1 is correct?

P11 I7-8: longwave and shortwave are equal
P12 I36-37: The SHL is measured in between 925 and 700hPa, not at the surface. Do the authors conclude there is a cooling of the SHL intensity due to the water vapor?

Figures : For all the figures, please add the caption under the figures

Fig 1: Not relevant since the study is only for June 2011. Please provide the same map during the campaign and some information relevant to characterize 2011 vs the climatology.

Fig 6: This figure is not readable. Please change to a scatter plot, more adapted.

Fig.9: I recommend to transform some panels in scatter plots

Fig. 11: I recommend to remove this figure.

Fig 17: Please modify this figure. It is not readable.

---

## Referee Comment (RC1) · Anonymous Referee #1 · 27 Jul 2017

This paper used field experiment data at BBM in southern Algeria from June 2011 and a radiative transfer model to calculate the effects of dust and water vapor on radiation budget both at the surface and the TOA in order to understand the radiative processes within the SHL during summer. Generally, the manuscript is straightforward and well organized. However my main concern is that some of the input data for the RT model may cause large uncertainties that are helpless to fill the research gaps as the authors mentioned in the introduction. For example, dust can absorb thermal infrared radiation, the night time AOD estimated from the nephelometer, which measures aerosol extinction coefficient near the surface, could induce a large error without an accurate aerosol extinction profile. Reanalysis data generally has poor representations of clouds and

their properties. However, the authors selected clouds properties from the reanalysis. These could directly affect the reliability of the model results.

Sections 2 and 3 are a bit long. I would recommend to combine and simplify this part. What the authors concluded cannot be totally supported only from the radiative forcing and heating rate calculations.

The manuscript also need a thorough editing. Some typos and confusing expression make the text difficult to follow at times.

---

## Referee Comment (RC2) · Anonymous Referee #2 · 5 Oct 2017

Review of "The summertime Saharan heat low: Sensitivity of the radiation budget and atmospheric heating to water vapor and dust aerosol." by Alamirew et al.

This paper explores the sensitivity of the radiation budget within the Sharan Heat Low (SHL) to changes in water vapor and dust in order to understand the influences of each on synoptic (and potentially longer) time scales. This sensitivity analysis is carried out using observations during the intensive measurement period of the Fennec experiment during June 2011 at Bordj Badji Mokhtar in central Algeria. The main finding presented here is that dust and water vapor contribute approximately equally to variability of the SHL radiative budget.

[Figure]

I have some concerns with regards to the tuning of the dust in the RT model (although I just may be misunderstanding the description of the model setup). I think the authors should do more in terms of carrying their error analysis throughout the entirety of the dust and water vapor forcing analysis, and I think the paper is way too long (unnecessarily wordy and too many superfluous plots/tables). I am suggesting a major revision, but the work required to satisfy these comments is super "do-able".

Major Comments:

1. Error Analysis: The authors spend a good bit of time estimating uncertainty in their modeled fluxes via comparison to satellite retrieved fluxes. However, when it comes to the data analysis, these uncertainties are not taken into consideration. I think it's great that the authors have a handle on the RT model errors, but I think it would be far more useful to carry those uncertainties throughout the entirety of Section 4. Doing so would make the paper and results much stronger and would afford the community opportunity to make a more precise comparison between yours and future dust forcing estimates.

2. Radiative Transfer Model. To generate the mie coefficients the authors use two different size distributions (Dubovik and Ryder) but the same index of refraction. However, what's the source of the refractive index? The authors conclude that the Dubovik size distribution is more representative of the actual size distribution based on a comparison of the model and observed/retrieved fluxes. However, it is completely possible that the index of refraction used here also biased. For example, it's possible that the Ryder distribution is correct but doesn't produce enough SW dust forcing because the MEC is too low at the appropriate size parameter, thus the forcing in the SWE for Dubovik would better match observations because it's biased towards smaller particles. At any rate, my only point is that you have two degrees of freedom and you can't say conclusively that one size distribution is more representative than another one b/c the index of refraction isn't constrained.

3. RT Model: The authors state that the vertical profile of the dust mass mixing ratio is

adjusted so that for a given MEC the AOD matches observations. Is the profile linearly scaled by a single value to match the observations? Is a single coefficient derived for all cases or is this done independently for each RT simulation?

4. Flux comparisons: It the text it is not clear if the flux comparisons are performed in a robust manner. For example, why are monthly mean fluxes from CERES compared to the observations and output from the model? The proper way to conduct the comparison with CERES would be to access the daily nighttime and daytime data and then subsample the observations/RT model output/GERB retrievals in order to conduct an apples-to-apples comparison. The authors acknowledge this (Page 9 line 35) so it's puzzling why a more thorough analysis wasn't performed. This approach includes the task of making comparisons to the reanalysis data (again, authors note that interpolating MERRA surface temperature3 may be biasing the flux comparisons). Furthermore, more insight would likely be gained by comparing the clear-sky fluxes only, since cloud forcing is not important to the study.

5. Flux comparisons: Tables and Figures. There are too many tables and the main figure (9) for this section is not particularly useful. Firstly, the tables are cumbersome and don't communicate the main results well (for example, color could be used to indicate if RT model output or reanalysis output is biased high or low in comparison to surface obs or satellite retrievals. In addition, the flux comparison Fig 9 are tough to interpret because the annual cycle is included. A better way to do this is to have one plot comparing the mean annual cycles, and another comparing the anomalies.

6. Forcing Efficiencies: The efficiencies reported for dust and IWV should also include the associated 95% confidence intervals from the linear regression.

7. Figures 12 and 16 aren't really all that interesting. Consider including observations here as well (at least for TOA). BTW - CERES produces surface flux products. These could be folded into the analysis as well.

8. Figure 17 is impossible to read/interpret, and I don't even wear glasses (yet)! Please

consider a more simple and straightforward way to describe the vertical sensitivities. A good rule-of-thumb would be to only include in the plot information that you actually describe in the text.

Minor Comments: 1. Figures: The individual panels of the figures should be labeled (a., b., c., . . .).

2. Figure 5: This figure is not very useful in terms of understanding the relationship between the AODs and IWV. Can you please just replace with one or two scatter plots?

3. Figure 6. If the authors removed the diurnal cycle from this plot we'd have an easier time interpreting the magnitude of the biases. As it is presented here, the magnitude of the differences are small relative to the magnitude of the diurnal temperature changes, making it difficult to interpret the results.

4. Page 9, Line 2: You write "Dubovik Optical Properties" do you mean optical properties generated using the size distribution from Dubovik and the index of refraction that you've been using up to now (that hasn't been referenced)? It's just not clear.

5. Page 13, Paragraph starting on line 28: The finding that IWV and dad contribute approximately equally to variance in the radiative budget is by far the most interesting (and new) finding reported in the paper. Why not take a little more space to flesh this out a bit? And please include the uncertainty estimates.

---

## Author Comment (AC4) · 30 Oct 2017

1. This paper used field experiment data at BBM in southern Algeria from June 2011 and a radiative transfer model to calculate the effects of dust and water vapor on radiation budget both at the surface and the TOA in order to understand the radiative processes within the SHL during summer. Generally, the manuscript is straightforward and well organized. However my main concern is that some of the input data for the RT model may cause large uncertainties that are helpless to fill the research gaps as the authors mentioned in the introduction.

Response

We fully recognise the challenge of adequately constraining the input data to the RT model in this region, where observations are sparse and as a results reanalyses models have limited assimilation of observations. This is indeed a challenge and one which the Fennec project set out to address. In using Fennec data we therefore utilise the best available data for our RT simulations. Moreover, we undertake a very comprehensive analysis of the sensitivity of radiative heating to uncertainties in those input field not directly measured during Fennec. Indeed reviewer 1 felt that this model configuration section was too comprehensive to be included in the main paper! So we believe we have addressed the issue of data input uncertainty as thorough and comprehensive manner as could be reasonably expected. This is now included in the supplementary material section so as not to distract from the core hypotheses the paper sets out the test.

2. For example, dust can absorb thermal infrared radiation, the night time AOD estimated from the nephelometer, which measures aerosol extinction coefficient near the surface, could induce a large error without an accurate aerosol extinction profile.

Response

Lack of complete input data is one of the challenges in the study of radiative effect of aerosols. Because of this, there is always assumptions or approximations to overcome the arising difficulties. Using surface nephelometer measurements to estimate night time AOD will not significantly affect our result. This is because there is only LW forcing at night which is in general smaller compared with SW forcing. Besides researchers practically use uniform dust extinction profile across the boundary layer as the difference in forcing results compared with the actual extinction profile is not small. [Liao and Seinfeld 1998, Osipov et al., 2015,] We have also confirmed this through a sensitivity experiment to test the difference in LW radiative flux and heating rate when we use different daytime and nighttime extinction profile. We find a small difference less than 3 W.m-2 both at the surface and TOA. The atmospheric heating rates do not change significantly when different extinction profiles are used for day and night except small

difference in the lower levels by less than 0.20 K day-1. We conclude in general that this will not affect what we wanted to show and hence the overall result of the paper.

3. Reanalysis data generally has poor representations of clouds and their properties. However, the authors selected clouds properties from the reanalysis. These could directly affect the reliability of the model results.

Response

This was also our concern at the beginning of this research work as we understand the limitations of cloud representations in models. We could have undertaken the RT experiments only in clear sky mode as many other authors choose to do. We do include clear sky only experiments but we complement these with all sky experiments to provide a more thorough and comprehensive analysis, from which we compare observations of TOA fluxes in which cloud screening is problematic. Our all sky RT experiments use what we feel is the best available 3-D information on cloud, that comes from the reanalysis models. Alternative cloud profiles for RT models simulations is not available. It is totally expected that our results will bring error due to cloud under (or mis) representation. We discuss this on Page 9: L14-20 of corrected draft and page 3:L1-5, L14-16 of supplementary material. However, we stand by our analysis not least because comparison of the errors in the all sky vs clear sky RT results actually provide some first order indication of the error on radiative budget due to underestimated cloud in reanalysis dataset. We have included a clearer and more explicit caveat regarding the limitations of the cloud fields in our experiments and note the need for further work in this area.

Changes Made

Page 3:L14-16 of supplementary material.

4. Sections 2 and 3 are a bit long. I would recommend to combine and simplify this part.

Response

This part has been restructured in a more clear way (please refer to the comment of reviewer 1, reviewer #1 Major Comment a #1.)

Changes Made

Refer to the response of reviewer #1 Major Comment a #1 for the simplified layout of the paper.

5. What the authors concluded cannot be totally supported only from the radiative forcing and heating rate calculations.

Response

Reviewer #1 also raised this comment. Please refer the responses made to reviewer #1, Major Comment C #2

6. The manuscript also need a thorough editing. Some typos and confusing expression make the text difficult to follow at times.

Response

Manuscript thoroughly read and corrections made to typos.

Please also note the supplement to this comment:
https://www.atmos-chem-phys-discuss.net/acp-2017-397/acp-2017-397-AC4-supplement.pdf

**Supplement:**

Supplementary of

**The summertime Saharan heat low: Sensitivity of the radiation budget and atmospheric heating to water vapor and dust aerosol.**

5  Netsanet K. Alamirew[1*], Martin C. Todd[1*], Claire L. Ryder[2], John H. Marsham[3], Yi Wang[1]

[1]Department of Geography, University of Sussex, Brighton, UK

[2]Department of Meteorology, University of Reading, Reading, UK

[3]School of Earth and Environment, University of Leeds, Leeds, UK

[*]Correspondence to: N. Alamirew na286@sussex.ac.uk or M. Todd m.todd@sussex.ac.uk

10

15

20

25

**SUPPLEMENTARY MATERIAL**

**S1 Optical Properties of dust**

SSA values in the band covering the spectral range 0.32 to 0.69 μm are 0.82 and 0.91 for Fennec-Ryder and Dubovik respectively. The coarser particles in Fennec-Ryder result in a lower SSA – i.e. more absorbing dust. Note that in the model since AOD is fixed based on the observed AOD, the vertical profile of dust mass mixing ratio is adjusted so that when combined with the MEC shown in fig. 5 (main paper), the AOD in the spectral range 0.32 to 0.69 μm is correct. Therefore the differences in MEC between the two datasets shown in fig. 5 cannot result in differences within the RT model. However, differences in SSA and g are able to exert different impacts on the radiative fluxes within the RT model, as described in section 4.

**S2 RT Sensitivity experiments to choice of inputs**

For some quantities, we do not have direct observations so we use alternative data from various sources. In the 'configuration mode' we test the uncertainty of the modelled radiative fluxes to uncertainties in these model inputs using the experiments summarised in Table S1. Then comparison of TOA fluxes with satellite observation allows us to arrive at what we consider to be an 'optimal' model configuration for the subsequent model 'experiment mode' analysis.

**(i) Surface skin temperature.** Since there are no complete observations of skin temperature we use reanalysis products as inputs to the RT code and we use both these data to further investigate sensitivity of flux to uncertainty in skin temperature. Figure 6 displays the time series of surface skin temperature from ERA-I, MERRA, and CERES footprint data. Root mean square difference (RMSD) of the reanalysis products with respect to CERES-footprint data are high (4.5 K and 5.5 K for MERRA and ERA-I, respectively). Fig. S1 further confirms that MERRA has strong diurnal anomalies compared with ERA-I. Despite the higher RMSD of ERA-I skin temperature compared with RMSD of MERRA, the RMSD of ERA-I 2 m air temperature (Figure 6) with respect to flux tower measurement is 3.1 K (3.7 K, MERRA). The relatively bigger RMSD in skin temperature could be due to the uncertainty in CERES measurements.

**(ii) Surface emissivity.** We test the sensitivity of radiative fluxes to uncertainty in estimates of surface emissivity using CERES measurements and MERRA outputs that have monthly mean values of 0.89 and 0.94 respectively.

**(iii) Surface albedo.** We noted that in contrast to observations the reanalysis products have weak representation of the diurnal cycle in surface albedo (Figure 3). Although we use observed surface albedo throughout all our experiment model RT runs, we also test the sensitivity of TOA shortwave flux to reanalysis surface albedo errors.

**(iv) Dust Size Distribution.** We test sensitivity of dust radiative forcing to the two size distribution described in the paper, Dubovik dust and Fennec-Ryder dust. Comparison of TOA direct radiative effect of dust with measurements and previous published results are used as basis for the choice of size distribution of dust. The results of sensitivity of radiative flux to dust size distributions is provided in the paper.

**(iv) Cloud Properties.** Acquiring observations of the vertical structure of clouds of sufficient quality for radiative transfer calculations is always challenging. Here we use the ERA-I and MERRA outputs of cloud fraction, liquid and ice water mixing ratios. Cloud is treated to have maximum overlap in a column where ice

and water are mixed homogeneously. During the Fennec period, cloud was characterised by shallow cumulus or altocumulus near the top of the PBL and occasional deep convection. It is likely that the relatively coarse vertical and horizontal resolution of both reanalysis models will have considerable bias and we recognise that this is likely to underestimate the true cloud-related uncertainty, and for example, M16 suggest that ERA-I underestimate cloud fraction by a factor of 2.5.

**S3 RT model optimum configuration**

Sensitivity of RT simulated fluxes to uncertainty in the surface skin temperature and emissivity is low compared to the sensitivity to other factors (Table S1) with variations of ~2 W m$^{-2}$ at TOA and 5-6 W m$^{-2}$ at surface. Based on bias with respect to CERES-EBFA observations we use ERA-I skin temperature and MERRA emissivity products for the 'optimal' configuration.

TOA fluxes are not strongly sensitive to the choice of cloud properties with TOA net flux variations of ~4 W m$^{-2}$. On the basis of bias with respect to observations we select the ERA-I cloud properties. It should however be noted that reanalysis is likely to have problems resolving the type of cloud over Sahara i.e. shall, shallow top of boundary layer cumulus. It is interesting to note that TOA radiative fluxes are quite sensitive to the errors in surface albedo from reanalysis with differences up to 16 W m-2 compared to the optimum configuration, which used observed surface albedo. This suggests that it is important to have good observational data, which captures the strong diurnal cycle of surface albedo to achieve accurate radiative fluxes.

**Table S1**. Summary of model configuration mode sensitivity analysis

| Sensitivity input variable | Source of data for sensitivity run | Sensitivity results | 'Optimal configuration choice |
|---|---|---|---|
| surface albedo | Fennec measured quantity versus ERA-I | Difference of upto 16 W m$^{-2}$ in TOA net SW flux | Surface Albedo calculated from surface flux measurements |
| skin temperature | ERA-I MERRA | Difference of 6 W m$^{-2}$ in surface net LW flux | ERA-I skin temperature estimate |
| Surface emissivity | CERES MERRA | Differences of 2.3 W m$^{-2}$ at TOA LW flux and 5 W m$^{-2}$ at the surface. | MERRA reanalysis estimates |
| Cloud fraction and mixing ratio | ERA-I MERRA | Difference of 4 W m$^{-2}$ both at TOA and surface net SW flux | ERA-I |
| dust size distribution | Dubovik FENNEC-Ryder | TOA SW dust DRE -2 W m$^{-2}$ Using Dubovik and 23 W m$^{-2}$ using Ryder-FENNEC | Dubovik |

[Figure]

Figure S1. Same as fig. 6 but for anomalies

[Figure]

Figure S2. Same as fig 9 but for anomalies

[Figure]

Figure S3 DRE due to Dust: time series of TOA shortwave (a), longwave (b), net (c) and surface shortwave (d) longwave (e) and net (f) wdnC-nDnC flux

---

## Author Response (AR1)

**'The summertime Saharan heat low: Sensitivity of the radiation budget and atmospheric heating to water vapour and dust aerosol' by Netsanet K. Alamirew et al**

**Response to referees and reviewer**

The comments and suggestions made by all referees and reviewer are useful. We have addressed the comments raised. Our responses and changes (if any) are indicated in the corrected version of the paper. For clarity we put original comment of the reviewer (typed in italic font) followed by our responses to make it easy to follow.

**Response to interactive discussion Short Comment (SC) from C. Lavaysse**

**Major Comment a.**

1. *Section 3 is not clear. Quite complicated to understand all the configurations and the conclusions drawn from these results on the choice of certain parameters. Finally choices are not really justified and I am not sure if it is necessary to provide all the information. I would recommend to simplify this section and to put some results in the supplementary material.*

   **Response**

   Part of section 3 has been moved to the supplementary material (Section S2). This includes all the model configuration analysis. Accordingly, Section 3 now describes the data and the design of the hypothesis testing experiments and Section 4 focuses only on the results of those experiments.

   **Changes Made**

   We have reorganized section 2 and 3 into a more clear structure. The new structure of the whole paper is as follows.

   Section 1. Introduction
   Section 2. Description of RT model
   Section 3. Data and method
       3.1. Observed top of atmosphere and surface radiation measurements
       3.2. Atmospheric profile and surface characteristics
       3.3. Dust properties and extinction profile
       3.4. RT model Experiments
   Section 4. Results and discussions.
       4.1. RT model validation
       4.2. The radiative flux and heating effects of dust and water vapour
           4.2.1. Dust

4.2.2. Water vapour

4.2.3. The relative effects of dust versus water vapour

Section 5. Summary and Conclusions

Original draft Page 6:L19-40, Page 7:L1-3, Page 7:L28-37 moved to supplementary material (section S2). See also minor comment #3.

2.   *In this section I also found some parts not clear: p5 l5-12; it is quite weird to compare observations assimilated with model dataset? The authors do not explain the remaining errors. Is it due to the assimilation procedure?*

**Response**

We are pointing the fact that despite assimilation of the radiosonde data there remain biases in the reanalysis. Fennec was a short-term experiment and since then there remains only one radiosonde station for the whole Sahara. As such, the reanalysis errors we derive are almost certainly much lower than those typical of the rest of the Sahara. We also now cite the errors estimated from Garcia-Carreras who compared radiosonde data to a forecast model first guess (independent of assimilation)

The magnitude of errors are different among the different reanalysis products. The possible reasons for the remaining error between observation and reanalysis products could be due to differences in models core dynamics and in assimilation procedures.

**Changes Made**

Corrected draft Page 4. L36-38. A statement added suggesting the possible reasons for differences in error among reanalyses.

**Major Comment b.**

1.   *Section 4 is too descriptive with too much information that are not necessarily significant or important to the conclusions of this study. This is particularly true p9 and 10. I strongly recommend to reduce this part to the most important results and to put the others results into an annex.*

**Response**

Part of section 4 has been moved to the supplementary Material (section S3), specifically sections describing the sensitivity experiments towards the model optimum configuration, as we agree these are not the key significant results.

We choose to retain some of the results originally presented in pages 9-10 because we feel it is important to demonstrate that the simulated quantities of top of atmosphere radiation budgets are within the observational uncertainties. To give sense of results in subsequent sections, it is necessary to have a feeling of the surface and TOA radiative budget under the mean state.

**Changes Made**

Original draft page 8:L30-33, page 9:L3-8 moved to supplementary material (section S3)

2.  *The summary of the subsection 4.1 is too speculative. How the authors can conclude the simulated flux errors of the optimal configuration are comparable to the observational uncertainties? What does 'acceptable' mean?*

**Response**

Given that we do not have accurate data for all the input required to run the RT model, it is not unexpected to get some uncertainty in our results. However we have chosen the inputs in such a way that the calculated flux are as close as possible to observation. This is what we mean by an 'optimum' model configuration. The optimum configuration is deemed to be 'acceptable' because the model error in top of atmosphere fluxes (perhaps the single most important quantity) with respect to observations is within the uncertainty in the observational estimates of those quantities. Model estimates lying within observation range is a commonly used indicator of acceptable model performance. Thus we suggested the RT model is configured to produce acceptable results and thus can be used for further experiments.

**Major Comment c.**

1.  *Some conclusions are too speculative. The authors conclude about the impacts of the dust aerosols and water vapor on the SHL but, in that study, only June 2011 is used. The SHL is the most important from end of June to mid of September (when it is installed in its Saharan location). Even if the authors used only one month (June), they have to characterize this specific year to the climatology (in term of dust, humidity, large scale forcings). This point concerns the title ('summertime' is not appropriate), the conclusions (p15 l8-10), and the abstract.*

**Response**

We agree that the period of study does not coincide with the peak of the summer season when the SHL is established in its northernmost position. However, we are limited by the period of the Fennec field campaign whose data underpin our analysis. Accordingly we have changed all references to 'summertime' to 'early summer'. In addition, in Section 3.2 we note that during our study period of June 2011 the SHL underwent a rapid transition from a 'maritime phase' to a 'heat low' phase. As such our analysis actually covers the transition period and SH states characteristic of both early and high summer. We have now amended this section to include an analysis of the conditions during June 2011 with respect to the mean conditions during June.

**Changes Made**

References to summer changed to summertime.

Figure 1 changed to show position of SHL in June, 2011.

Corrected draft Page 16:L14-20. A paragraph added

2. *Also the discussion on the impacts on the SHL pulsations should be carefully discussed since the authors do not analyze the contribution of the large scale temperature advections and they never show the real position of the SHL in June 2011 (in June, the SHL is migrating to the north with a large spatial variability).*

**Response**

Real position of SHL in June is shown in Fig 1.

The comments on our reference to variability in SHL specifically the 'pulsating' of SHL intensity and the potential role of dust and water vapour feedbacks in this process is also raised by anonymous referee #1. We do feel it is important in this paper to relate the radiative heating rates derived from our RT simulations to the behaviour of the SHL, but of course recognise that the full dynamical response requires an analysis of advective heating. As such in the original paper p16 para 1 we note that radiative heating is of 'comparable magnitude' to published estimates of advective cooling from comparable monsoon surge type events. In this way we make only a broad inference about the net effects of advective and radiative terms on the SHL. We have now changed the text slightly to emphasise the speculative nature of this inference.

**Changes Made**

Corrected draft Page 15:L26-28. Additional statement included.

'

3. *Finally at climatological scale, the authors should pay attention to the climatological evolution of the dust that tends to reduce (p15 l16)*

**Response**

Our comment in the original draft page 15:L16 concerns other analysis which implicate long term trends in SHL temperature to that in WV, but do not include dust in their analyses. We simply aimed to point out that this should not be neglected. Our paper is not concerned with resolving long term trends in dust over the SHL so we do not include plots of long term satellite derived AOD over the SHL.

**Major Comment d.**

1. *Some figures are not readable*

   **Response**

   Unreadable figures corrected.
* * *
**Minor Comments**

1. *P2 l11 the authors should mention this reference: Lavaysse, C., Flamant, C., Evan, A. et al. Clim Dyn (2016) 47: 3479. doi:10.1007/s00382-015-2847-z*

   **Response:** Reference included, P2:L11 and reference section page 18: L32.

2. *P6 l4; the two phases mentioned are not so clear.*

   **Response:** These two phases are previously stated on original draft page 4:L40 and page 5:L1

3. *P6 l19: title of subsection 3.2 not clear, please rephrase*

   **Response:** changed to 'RT sensitivity experiments to choice of inputs', now moved to supplementary material.

4. *P6 l24: optimal to what?*

   **Response:** Optimal configuration means model configured to produce results closest to observations.

5. *P6 l37-38; how do the authors conclude the Ceres measurements are uncertain and that explain the large RMSE? The term RMSE refers to a reference (usually observations) that are considered as the correct value. Here, I do not understand what is the reference and how they can conclude that. Please clarify. Also the term RMSD (difference) should be more appropriate.*

   **Response:** We do agree with the reviewer's comment that RMSD is comparison of modeled versus observation. From the data we have CERES is considered correct, despite its limitations as with any observation, can be used to measure the error modelled variables.

   **Changes Made:** RMSE changed to RMSD in all occasions.

6. *P6 l39-40: the authors provide some results without explanations, what are these results (mean =...) and please clarify the conclusions/interest of this point?*

   **Response:** Rephrased, point of interest described in section 5

7. *P7 subsection 3.2.2 I recommend to put the first part of the paragraph in the introduction section and the result in supplementary material.*

   **Response:** Some of the information and results on optical properties of dust is now moved to section S1 of supplementary material.

8. *P8 l1: Section 4.1 is correct?*

   **Response:** Corrected

9. *P8 l11: Is it necessary to use this acronym?*

   **Response:** Acronym definitions summarized in table 2. To be consistent throughout the paper, we found it necessary to use acronym.

10. *P8 l27: Section 3.1 is correct?*

**Response:** corrected, for the details look at response to Major comment a.

11. *P11 l7-8: longwave and shortwave are equal*

**Response:** TOA SW DRE of dust is small, whereas LW has a net warming effect at TOA(less LW escaping out of atmosphere due to dust.)

12. *P12 l36-37: The SHL is measured in between 925 and 700hPa, not at the surface. Do the authors conclude there is a cooling of the SHL intensity due to the water vapor?*

**Response:** Here we are discussing the immediate radiative effect of dust and water vapour. But the net effect may not be cooling as the feedback resulting from surface warming in the LW and thus more sensible heat flux could result in net warming of the atmosphere which needs further investigation using regional climate models that include the feedback processes.

13. *Figures : For all the figures, please add the caption under the figures*

**Response:** All changes are made to the figures according to the given recommendation.

**Response to Referee Comment (RC) from Anonymous Referee #1**

1. *This paper used field experiment data at BBM in southern Algeria from June 2011 and a radiative transfer model to calculate the effects of dust and water vapor on radiation budget both at the surface and the TOA in order to understand the radiative processes within the SHL during summer. Generally, the manuscript is straightforward and well organized. However my main concern is that some of the input data for the RT model may cause large uncertainties that are helpless to fill the research gaps as the authors mentioned in the introduction.*

   **Response**

   We fully recognise the challenge of adequately constraining the input data to the RT model in this region, where observations are sparse and as a results reanalyses models have limited assimilation of observations. This is indeed a challenge and one which the Fennec project set out to address. In using Fennec data we therefore utilise the best available data for our RT simulations. Moreover, we undertake a very comprehensive analysis of the sensitivity of radiative heating to uncertainties in those input field not directly measured during Fennec. Indeed reviewer 1 felt that this model configuration section was too comprehensive to be included in the main paper! So we believe we have addressed the issue of data input uncertainty as thorough and comprehensive manner as could be reasonably expected. This is now included in the supplementary material section so as not to distract from the core hypotheses the paper sets out the test.

2. *For example, dust can absorb thermal infrared radiation, the night time AOD estimated from the nephelometer, which measures aerosol extinction coefficient near the surface, could induce a large error without an accurate aerosol extinction profile.*

   **Response**

Lack of complete input data is one of the challenges in the study of radiative effect of aerosols. Because of this, there is always assumptions or approximations to overcome the arising difficulties. Using surface nephelometer measurements to estimate night time AOD will not significantly affect our result. This is because there is only LW forcing at night which is in general smaller compared with SW forcing. Besides researchers practically use uniform dust extinction profile across the boundary layer as the difference in forcing results compared with the actual extinction profile is not small. [Liao and Seinfeld 1998, Osipov et al., 2015,]

We have also confirmed this through a sensitivity experiment to test the difference in LW radiative flux and heating rate when we use different daytime and nighttime extinction profile. We find a small difference less than 3 W.m$^{-2}$ both at the surface and TOA. The atmospheric heating rates do not change significantly when different extinction profiles are used for day and night except small difference in the lower levels by less than 0.20 K day$^{-1}$. We conclude in general that this will not affect what we wanted to show and hence the overall result of the paper.

3. *Reanalysis data generally has poor representations of clouds and their properties. However, the authors selected clouds properties from the reanalysis. These could directly affect the reliability of the model results.*

**Response**

This was also our concern at the beginning of this research work as we understand the limitations of cloud representations in models. We could have undertaken the RT experiments only in clear sky mode as many other authors choose to do. We do include clear sky only experiments but we complement these with all sky experiments to provide a more thorough and comprehensive analysis, from which we compare observations of TOA fluxes in which cloud screening is problematic. Our all sky RT experiments use what we feel is the best available 3-D information on cloud, that comes from the reanalysis models. Alternative cloud profiles for RT models simulations is not available. It is totally expected that our results will bring error due to cloud under (or mis) representation. We discuss this on Page 9: L14-20 of corrected draft and page 3:L1-5, L14-16 of supplementary material. However, we stand by our analysis not least because comparison of the errors in the all sky vs clear sky RT results actually provide some first order indication of the error on radiative budget due to underestimated cloud in reanalysis dataset. We have included a clearer and more explicit caveat regarding the limitations of the cloud fields in our experiments and note the need for further work in this area.

**Changes Made**

Page 3:L14-16 of supplementary material.

4. *Sections 2 and 3 are a bit long. I would recommend to combine and simplify this part.*

**Response**

This part has been restructured in a more clear way (please refer to the comment of reviewer 1, reviewer #1 Major Comment a #1.)

**Changes Made**

Refer to the response of reviewer #1 Major Comment a #1 for the simplified layout of the paper.

5. *What the authors concluded cannot be totally supported only from the radiative forcing and heating rate calculations.*

**Response**

Reviewer #1 also raised this comment. Please refer the responses made to reviewer #1, Major Comment C #2

6. *The manuscript also need a thorough editing. Some typos and confusing expression make the text difficult to follow at times.*

**Response**

Manuscript thoroughly read and corrections made to typos.

**Response to Referee Comment (RC) from Anonymous Referee #2**

**Major Comments**

1. *Error Analysis: The authors spend a good bit of time estimating uncertainty in their modeled fluxes via comparison to satellite retrieved fluxes. However, when it comes to the data analysis, these uncertainties are not taken into consideration. I think it's great that the authors have a handle on the RT model errors, but I think it would be far more useful to carry those uncertainties throughout the entirety of Section 4. Doing so would make the paper and results much stronger and would afford the community opportunity to make a more precise comparison between yours and future dust forcing estimates.*

**Response**

We agree the importance of including error analysis despite we have reduced the uncertainty using

sensitivity experiments. This is addressed qualitatively to some extent in section 4, i.e. error associated with the uncertainties in the input.

**Changes made**

Additional information quantitatively expressing the error in flux calculation associated with uncertainties in some of the input data is provided. Page 8 L25-28 and L37-39.

2. *Radiative Transfer Model. To generate the mie coefficients the authors use two different size distributions (Dubovik and Ryder) but the same index of refraction. However, what's the source of the refractive index? The authors conclude that the Dubovik size distribution is more representative of the actual size distribution based on a comparison of the model and observed/retrieved fluxes. However, it is completely possible that the index of refraction used here also biased. For example, it's possible that the Ryder distribution is correct but doesn't produce enough SW dust forcing because the MEC is too low at the appropriate size parameter, thus the forcing in the SWE for Dubovik would better match observations because it's biased towards smaller particles. At any rate, my only point is that you have two degrees of freedom and you can't say conclusively that one size distribution is more representative than another one b/c the index of refraction isn't constrained.*

**Response**

We agree that the refractive index may cause uncertainty in the flux calculations especially in the SW absorption. It is also interesting to test the sensitivity of radiative flux to refractive index. In general for a given size distribution of dust, when refractive index is increased net SW heating will increase and net LW cooling will increase to a lesser extent. This however is a complicated function depending on the surface albedo and cloud. (Liao et al., 1998). Here we used recent measurements for dust refractive index over the Sahara (Ryder et al., 2013) which is function of the composition of dust particles, independent of the size distribution. It could be possible that if we reduce the refractive index, the SW heating will reduce in Ryder distribution, which is the biggest discrepancy compared with satellite measurement. But we haven't made sensitivity test as we have measured refractive index.

3. *RT Model: The authors state that the vertical profile of the dust mass mixing ratio is adjusted so that for a given MEC the AOD matches observations. Is the profile linearly scaled by a single value to match the observations? Is a single coefficient derived for all cases or is this done independently for each RT simulation?*

**Response**

To be clearer, first an average extinction profile is derived from CALLIOP and this profile is used to derive the extinction profile at each time step, i.e. the average profile is adjusted to match the measured

AOD from AERONET. So to answer the question, for each RT calculations independent extinction profile is derived.

4. *Flux comparisons: It the text it is not clear if the flux comparisons are performed in a robust manner. For example, why are monthly mean fluxes from CERES compared to the observations and output from the model? The proper way to conduct the comparison with CERES would be to access the daily nighttime and daytime data and then sub sample the observations/RT model output/GERB retrievals in order to conduct an apples-to-apples comparison. The authors acknowledge this (Page 9 line 35) so it's puzzling why a more thorough analysis wasn't performed. This approach includes the task of making comparisons to the reanalysis data (again, authors note that interpolating MERRA surface temperature may be biasing the flux comparisons). Furthermore, more insight would likely be gained by comparing the clear-sky fluxes only, since cloud forcing is not important to the study.*

**Response**

An important aspect of this study that needs to be noted is it is intended to provide season (one month) study of the radiative budget and sensitivities to water vapour and dust variability over the Saharan heat low In order to do so we have used the best available input dataset through sensitivity experiments. It is useful to carry out comparison of the radiative flux at the time steps of CERES data (which is twice per day) as the referee suggested. We have actually made comparison of RT model outputs with CERES data with the respective time step to derive RMSE. This is presented on page 9 line 21(corrected draft). This will give us a good picture of the uncertainties of model simulations. However further comparisons using average of two time steps per day will not enable us to achieve the target we put at the outset.

To compare simulated flux with observation, GERB data is used. Further reanalysis data is also used which is available daily and thus used the same days as the RT model simulation days. CERES data is not used to compare simulated flux except for sensitivity experiments and estimate cloud DRE. We understand that using month mean CERES clear sky and all sky flux will bring some error but it will give us first order estimate of cloud DRE over the region. This will help emphasize need to improve the error on the radiative budget due to underestimated cloud in reanalysis dataset despite the challenges in making these comparisons.

5. *Flux comparisons: Tables and Figures. There are too many tables and the main figure (9) for this section is not particularly useful. Firstly, the tables are cumbersome and don't communicate the main results well (for example, color could be used to indicate if RT model output or reanalysis output is biased high or low in comparison to surface obs or satellite retrievals. In addition, the flux comparison Fig 9 are tough to interpret because the annual cycle is included. A better way to do this is to have one plot comparing the mean annual cycles, and another comparing the anomalies.*

**Response**

We agree to remove Table 5 since the information on this table is also found in Tables 2 and 3(corrected draft). An additional table is moved to the supplementary material.

Colours included on the wDwC results in tables 3 and 4(corrected draft Tables 2 and 3) red indicating model results overestimated and blue indicating model results underestimated compared with observation.

Some of the figures were corrected based on referee #1 and reviewer's comments. Figure 9(also Figure 6) is corrected and it is easier to read. We therefore keep it as it is. But have also made additional plot using anomalies but we put it in the supplementary document. See also page 9 L18-20.

**Changes made**

Table 5 removed
Colours used on column 6 of table 2 and table 3
Additional figure included in supplementary material page 3, figure S2

6. *Forcing efficiencies for dust and CIWV should also include the 95% confident interval from linear regressions.*

**Response**

We agree that the regressions should be expressed to 95% confidence level. All the regression results are expressed within the 95% confidence interval.

**Changes made**

These are included in section 4.2.1(page 10-11) and section 4.2.2(page 12) on the corrected draft.

7. *Figure 12 and 16 are not interesting. Consider including observations here as well (at least for TOA). BTW - CERES produces surface flux products. These could be folded into the analysis as well.*

**Response**

Here the plots are made using daily averaged variations in dust AOD or water vapour. That is dust AOD (and CIWV) is increased linearly in each RT run. This is a theoretical work designed to investigate the sensitivity of dust and water vapour on the radiation budget. There is no such observational data, at least at one particular point which is the observational data we used here. This

can be tested for a number of grid points of Satellite observation to see sensitivity of radiation to AOD variation (e.g Young et al., 2009). However this is not the objective of this study and thus it is not included.

8. *Figure 17 is impossible to read/interpret, and I don't even wear glasses (yet)! Please consider a more simple and straightforward way to describe the vertical sensitivities. A good rule-of-thumb would be to only include in the plot information that you actually describe in the text.*

**Response**

Figure made easier to read. Additional explanation regarding the figures provided

**Changes made**

Now we put the two panels of figure 17 as independent plots, Figure 16 and Figure 17 in the corrected draft.
Statement added on page 13, L18-19.

**Minor Comments**

1. *Individual panels of the figures should be labeled as a, b, c, ...*

**Response**

All figures prepared accordingly

2. *Figure 5: This figure is not very useful in terms of understanding the relationship between the AODs and IWV. Can you please just replace with one or two scatter plots?*

**Response**

We used SEVIRI AOD to show that there are cases where AOD is missed in AERONET which we suggest to be due to confusing dust with cloud. This we believe is important to show there are cases where dust might be missed in AERONET. We have complemented this using nephelometer measurements.

3. *Figure 6. If the authors removed the diurnal cycle from this plot we'd have an easier time interpreting the magnitude of the biases. As it is presented here, the magnitude of the differences are small relative to the magnitude of the diurnal temperature changes, making it difficult to interpret the results.*

**Response**

Figure 6 is now made easier to read and thus we keep it as it is. In addition we put the anomalies in the supplementary material. Additional information included in the supplementary material SP2 L:21-22

4. *Page 9, Line 2: You write "Dubovik Optical Properties" do you mean optical properties generated using the size distribution from Dubovik and the index of refraction that you've been using up to now (that hasn't been referenced)? It's just not clear.*

**Response**

Restated. Now on page 7 line 10 and 14.

Refractive index used comes from measurement. It is now made clear, Citation included, page 5 line 36

5. *Page 12, Paragraph starting on line 28: The finding that IWV and AOD contribute approximately equally to variance in the radiative budget is by far the most interesting (and new) finding reported in the paper. Why not take a little more space to flesh this out a bit? And please include the uncertainty estimates.*

**Response**

We agree this is an important point. Additional statement highlighting the significance of dust on controlling the radiative budget is included. Page 13 Line 4 of corrected draft.

**The Early summertime Saharan heat low: Sensitivity of the radiation budget and atmospheric heating to water vapor and dust aerosol.**

Netsanet K. Alamirew[1*], Martin C. Todd[1*], C.L. Ryder[2], John HM. Marsham[3], Y. Wang[1]

[1]Department of Geography, University of Sussex

[2]Department of Meteorology, University of Reading

[revised manuscript text omitted]

We undertake two types of RT experiment in this study:

(i) Model 'configuration mode' through which we test the sensitivity of simulated radiative fluxes to uncertainty in as many of the input variables as possible (see supplementary material Section S2, as described in Section 3.2, summarised in Table 1). The description and results of all sensitivity experiments to choice of different input data are provided in the supplementary material (Section S3 and Table S1). Here we present the results of the sensitivity experiments to dust size distribution since it is an important part of the paper. Sensitivity to the two contrasting dust size distributions is pronounced. As expected results using Fennec-Ryder dust show much stronger absorption in the shortwave compared with the Dubovik dust distribution, and the resulting TOA net shortwave flux is higher by 25 W m$^{-2}$ in the former. These shortwave fluxes using Fennec-Ryder distribution are not consistent with the GERB/CERES satellite observations (nor with previous estimates of shortwave DRE derived from satellite e.g. Yang et al. (2009); Ansell et al. (2014)) and we use dust optical properties generated using Dubovik size distribution in the optimum configuration. Recent work

suggests that the dust optical properties at BBM in June 2011 were significantly less absorbing than both those measured by the aircraft further west during Fennec, and the Dubovik representation (less absorbing, smaller sized) with SSA values of 0.99 (Rocha-Lima et al., submitted). Therefore, optical properties generated using Dubovik size distribution and measured refractive index represent intermediate values in terms of SW absorption.

Given that we don't have accurate data for all the input required to run the RT model, it is not unexpected to get some uncertainty in our results. However we have chosen the inputs in such a way that the calculated flux are as close as possible to observation. This will result in an acceptably configured model for experimental analysis presented next.

**(ii) Model 'experiment mode'** through which  we addressed the research questions, specifically to quantify the combined and separate DRE of water vapour and dust. To this end we undertook a number of experiments summarized in Table 1, with results described in Section 4. For all the experiments, RT calculations are made for each day using the atmospheric profiles at hourly time steps over the diurnal cycle, and the mean flux and heating rates are derived by averaging outputs at each time step. For this purpose, Aall input data are linearly interpolated to a one-hour temporal resolution.

For experiments with ('w') and without ('n') dust ('D') we simulate the $8^{th}$-$30^{th}$ June 2011 period. For sensitivity ('sen') experiments, we simulate linearly increased levels of dust AOD and water vapour. We use runs both with cloud ('C') and with no cloud (nC). For dust sensitivity experiment ('senDnC'), AOD is increased linearly over the range 0 (dust free) to 3 (extremely dusty), while keeping the mean value of water vapour constant. For water vapour sensitivity experiment ('senWVwDnC') the mean diurnal profile of water vapour is used but is scaled so that the column integrated water vapour increases from 10 to 40 kg m$^{-2}$ and the mean AOD is used in each case. The DRE for dust is derived by (i) subtracting TOA and surface fluxes of experiment wDnC from nDnC (ii) linear regression of the flux dependence on the range of dust AOD from the dust sensitivity experiments (senDnC), in which a single diurnal cycle is simulated. The impact of water vapour is determined by (i) composites of dry versus humid days from the nDnC experiment (ii) linear regression of the flux dependence on the range of water vapour from water vapour sensitivity experiments (senWVwDnC). The results of DRE of dust and water vapour are presented in Section 4.2.

[revised manuscript text omitted]

At the surface radiative flux is controlled much more strongly by dust than water vapour and with opposite sign: net cooling of -11 W m$^{-2}$ and warming of 6 W m$^{-2}$ per unit variability respectively. M16 find near zero warming from water (vapour and cloud). In our study the net effect of TOA versus surface is strong atmospheric warming of 18 W m$^{-2}$ per unit variability from dust and negligible warming (1 W m$^{-2}$ per unit variability) from water vapour. In contrast, M16 find almost equal warming from dust and water vapour (of 11-12 W m$^{-2}$ per unit variability). Although this radtiative transfer based analysis of the role of water vapour does not include  cloud that is implicitly included in M2016, we suggest that the co-variability of dust and water vapour hinders calculation of their independent effects in the observational analysis of M16.

In summary we find that dust and water vapour exert a similarly large control on TOA net radiation and therefore total column heating and by implication to the first order similar control on surface pressure in the SHL. However, the vertical structure of radiative heating from dust is far more complex than that for water vapour. The schematic, Fig. 16 and Fig. 17 illustrate the sensitivity of dust and water vapour respectively at different pressure levels. The grey shading in fig. 16 (fig. 17) represents amount of dust (water vapour) which  gives  AOD (CIWV) values shown on the horizontal axis when vertically summed. 
[revised manuscript text omitted]

We can therefore envisage an inherent tendency for pulsing in the SHL in which an intensifying SHL will lead towards monsoon surges, which act through near surface/low level radiative and advective cooling to weaken the SHL and through dust-radiative heating to stabilise the PBL, until dust deposition and export allow re-warming of the surface to re-invigorate the SHL.

Given the radiative effects described above the dynamical effects of dust variability on the low level convergence and mid-level divergence circulations will be greater than those of water vapour and require further model experiments to resolve. Whilst reanalysis models represent well the average radiative and heating effect of dust and water vapour they do not capture dust and water vapour variability well and the subsequent dynamical effects on the larger scale circulation.

The unique observations of the Fennec aircraft campaign suggested that fresh dust is much coarser than previously believed (Ryder et al., 2013b), with corresponding higher absorption, and this has significant impacts on the radiation budget (Kok et al., 2017). Our RT model simulations results suggest that such a dominant coarse mode is not consistent with TOA radiative flux observations at BBM. However, if dust is coarser than we assume here then the radiative effects of dust would be even stronger. Further observations on dust size distribution and optical properties are a priority requirement. In addition, further work should consider in much greater detail the radaitive effects of cloud based on detailed observations rather than the rather coarse estimates from reanalysis used here.

Our results showing the complex interplay of dust and water vapour on surface and PBL radiative heating stress the need for improved modelling of these processes over the SHL region to improve predictions including those for the WAM across timescales (e.g. Evan et al., 2015). Most models currently struggle in regard to short term variability in water vapour (Birch et al., 2014; Garcia-Carreras et al., 2013; Marsham et al., 2013a; Roberts et al., 2015), clouds (Roehrig et al., 2013; Stein et al., 2015) and dust (Evan et al., 2014), with many dust errors coming from moist convection ( Heinold et al., 2013; Marsham et al., 2011). Forecast models typically have mean biases of up to 2 kg m$^{-2}$ in column integrated water vapour (equivalent to change in 2.6 W m$^{-2}$ TOA net flux) and lack variability in dust, and so are expected to poorly represent these couplings. A focus on improved representation of advection of water vapour, clouds and convection in models should be a priority.

This paper has provided insight into the separate and combined roles of water vapour and dust in controlling the variability of the summertime radiative flux and heating rate over the SHL region. We recognise that generalising across all the SHL region for all summer months is problematic from one particular point and the short period of our study. Furthermore there still remains uncertainty in input dataset which includes surface characteristics and cloud. It is therefore necessary to have a more comprehensive dataset to reduce these uncertainties and thus improve quantitative results. Further research is thus necessary to confirm the results of our limited study spanning longer period of time and bigger domain.

[revised manuscript text omitted]
 (a, b, c) and shortwave (d, e, f). (a) and (d) mass extinction coefficient, (b) and (e) single scattering albedo, and (c) and (f) asymmetry parameter.  The continuous lines are the spectrally resolved optical properties and the horizontal lines are the band-averaged data that are used in the RT code.

**Figure 6**. CALIOP mean Extinction Coefficient profile at BBM 2006-13

**Figure 7**. AOD from AERONET and SEVIRI, and column integrated water vapour from FENNEC observation. Gray shades show driest days (11, 12, and 16), blue shades shows most humid days (18, 25, and 30), and green shade shows a major haboob event occurred on the 21[th] which resulted in large dust emission.

**Figure 6.** Surface skin temperature (SKT) and 2 m air temperature at BBM. Skin Temperature Black: ERAI, Red: MERRA, and Green Star: CERES footprint, 2 m air temperature Gold:ERAI and Cyan: Flux Tower measurement. The black and red stars denote ERAI and MERRA skin temperature at the time steps when there is CERES observation.

**Figure 7.** Wavelength dependence of optical properties of dust particle for longwave (top three panel) and shortwave (bottom three panel). The continuous lines are the spectrally resolved optical properties and the horizontal lines are the band-averaged data that are used in the RT code.

**Figure 8**. Mean Diurnal Cycle of TOA Flux. (a) shortwave and (b) longwave. Blue: SOCRATES results are from wDwC experiment. and green: GERB

**Figure 9**. Time series of Radiative Flux at BBM. TOA longwave (a), shortwave (b), and net (c). S(left column) and Surface (right column) shortwave (d), (SW), longwave (e), and net (f) longwave (LW), and net Radiative Flux at BBM. Black lines denote SOCRATES outputs, red line denote GERB measurements, green dots denote CERES measurementsThe bigger and red dots denote GERB measurements corresponding to CERES time steps.

**Figure 10**. Mean dDiurnal direct radiative effect of dust averaged for June 08-30, 20116. (a) TOA DRE of Dust (a) and (b) sSurface DRE of Dust (b). The bars show standard error over the diurnal cycle.

**Figure 11.** DRE due to Dust: time series of TOA shortwave (a), longwave (b), net (c) and surface shortwave (d) longwave (e) and net (f) wdnC nDnC flux

**Figure 121** Radiative budget as a function of dust AOD. Top row (a, b, c): TOA longwave (a), shortwave (b), and net (c). Second row (d, e, f): similar to top row but for surface. Third Row: atmospheric radiative convergence of longwave (g), shortwave (g), and net (i).

**Figure 123**. Mean rRadiative hHeating Rate Profile for June (08-30, 2011) at BBM. (a) Results from nDnC (dashed lines) and wDnC (solid lines) using FENEC profile and (b) MERRA Model output for all sky (solid lines) and clear sky (dashed lines) conditions. Blue, red, and green colours represent shortwave, longwave, and total heating rates respectively.

**Figure 134** Shortwave rRadiative hHeating rates (K.Day$^{-1}$) of dust in the atmosphere (DUST represents wDnC runs minusand CLEAN represents nDnC runs)

**Figure 145** Atmospheric hHeating rate profile for selected dry days, June 11, 12, and 16 (dashed lines) and moist days, June 18, 19, and 25 (solid lines)

**Figure 156** Same as FigureFig. 112 except for column integrated water vapour.

**Figure 167** Sensitivity of Radiative Flux (W.m$^{-2}$) to changes in dust AOD and column integrated water vapour. The numbers at each pressure level are downward shortwave (blue), longwave (red), and net (green) flux. The grey shade represents dust and water vapour amount in the atmosphere

**Figure 17** Same as Fig. 16 except for changes in column integrated water vapour

**Tables**

Table 1. Summary of model configuration mode sensitivity analysis

| Sensitivity input variable | Source of data for sensitivity run | Sensitivity results | 'Optimal configuration choice |
|---|---|---|---|
| | | | |

| surface albedo | Fennec measured quantity versus ERA-I | Difference of upto 16 $W.m^{-2}$ in TOA net SW flux | Surface Albedo calculated from surface flux measurements |
|---|---|---|---|
| skin temperature | ERA-I MERRA | Difference of 6 $W.m^{-2}$ in surface net LW flux | ERA-I skin temperature estimate |
| Surface emissivity | CERES MERRA | Differences of 2.3 $W.m^{-2}$ at TOA LW flux and 5 $W.m^{-2}$ at the surface. | MERRA reanalysis estimates |
| Cloud fraction and mixing ratio | ERA-I MERRA | Difference of 4 $W.m^{-2}$ both at TOA and surface net SW flux | ERA-I |
| dust size distribution | Dubovik FENNEC-Ryder | TOA SW dust DRE -2 $w.m^{-2}$ Using Dubovik and 23 $w.m^{-2}$ using Ryder-FENNEC | Dubovik |

Formatted Table

**Table 12**. Description of the RT 'experiment mode'. Names of different experiments acronyms are defined as 'n' = NO, 'w' = with, 'D' = Dust, 'C' = Cloud, 'WV' = water vapour, and 'sen' = sensitivity

| Name | Description | Water vapour | Aerosol | Cloud |
|---|---|---|---|---|
| nDnC | Dust free and Cloud free atmosphere | Observed 8[th]-30[th] June 2011 | None | None |
| nDwC | Dust free but cloudy atmosphere | Observed 8th-30th June 2011 diurnal cycle | None | ERAI MERRA |
| wDnC | Cloud free but dusty atmosphere | Observed 8th-30th June 2011 diurnal cycle | AERONET AOD scaled with CALIOP Extinction | None |
| wDwC | Dusty and Cloudy Atmosphere | Observed 8th-30th June 2011 diurnal cycle | AERONET AOD scaled with CALIOP Extinction | ERAI MERRA |
| senDnC | Sensitivity to full range of possible AOD | Mean diurnal WV | Linear increase in AOD 0.0 to 3.0 Constant AOD each time step for a given run | None |

| senWVwDnC | Sensitivity to full range of possible WV | Linear increase in TCWV from 10 to 40kg.m-2 at 2.5 kg.m$^{-2}$interval with mean diurnal WV profile | Mean Diurnal AOD | None |
| --- | --- | --- | --- | --- |

**Table 32.** Mean June 08-30, 2011 TOA Radiative flux at BBM (definition of acronyms are given in table 2). Values are in W.m$^{-2}$. The sign convention is that downward flux is considered as positive while upward flux is negative. On column 6 red (blue) fonts indicate model results overestimated (underestimated) compared with observation.

|  |  | nDnC | nDwC | wDnC | wDwC |
| --- | --- | --- | --- | --- | --- |
| TOA_SW | SOCRATES | 328 | 322 | 325 | **321** |
|  | GERB | -- | -- | -- | **314** |
|  | MERRA | 312 | 307 | 322 | **317** |
|  | ERAI | -- | -- | 336 | **324** |
| TOA_LW | SOCRATES | -313 | -304 | -297 | **-290** |
|  | GERB | -- | -- | -- | **-276** |
|  | MERRA | -314 | -- | -307 | **-296** |
|  | ERAI | -- | -- | -309 | **-294** |
|  |  |  | -- |  |  |
| TOA_NET | SOCRATES | 15 | 18 | 28 | **31** |
|  | GERB | -- | -- | -- | **38** |
|  | MERRA | -2 | -- | 15 | **20** |
|  | ERAI | -- | -- | 27 | **29** |
|  |  |  | -- |  |  |
|  |  |  | -- |  |  |

**Table 34.** Same as Table 3 but for surface radiative flux and observation from fennec instrument

|  |  | nDnC | nDwC | wDnC | wDwC |
| --- | --- | --- | --- | --- | --- |
| SRF_SW | SOCRATES | 237 | 232 | 192 | **187** |
|  | FENNEC_OBS | -- | -- | -- | **180** |
|  | MERRA | 220 | 215 | 190 | **185** |
|  | ERAI | -- | -- | 210 | **199** |
| SRF_LW | SOCRATES | -138 | -134 | -106 | **-103** |

| | | | | | |
|---|---|---|---|---|---|
| | FENNEC_OBS | -- | -- | -- | **-78** |
| | MERRA | -139 | -- | -119 | **-115** |
| | ERAI | -- | -- | -139 | **-132** |
| SRF_NET | SOCRATES | 99 | 98 | 86 | **84** |
| | FENNEC_OBS | -- | -- | -- | **103** |
| | MERRA | 82 | -- | 70 | **70** |
| | ERAI | -- | -- | 71 | **67** |

Table 45. TOA and Surface Direct Radiative Effect of Dust and Cloud

| | Dust DRE SOCRATES | | | Cloud DRE SOCRATES | | | Cloud DRE EBFA CERES EBFA | | |
|---|---|---|---|---|---|---|---|---|---|
| | SW | LW | NET | SW | LW | NET | SW | LW | NET |
| TOA | -3 | 16 | 13 | -4 | 7 | 3 | -15 | 16 | 1 |
| SURFACE | -45 | 32 | -13 | -5 | 3 | -2 | -19 | 11 | -8 |

Table 456. Sensitivity of Radiative Flux to Dust AOD and CIWV at selected altitudes. $SD^*$=Standard Deviation (0.8 for AOD and 5.5 g.kg$^{-1}$ for water vapour. Mean AOD = 1.2 and mean column integrated water vapour = 27.8 Kg.m$^{-2}$)

| Change in Flux | | SW | LW | NET |
|---|---|---|---|---|
| per unit AOD (W.m$^{-2}$) | TOA | -1.8 | 10.0 | 8.2 |
| | Surface | -33.8 | 19.8 | -14.0 |
| | Convergence | 32.1 | -9.7 | 22.4 |
| per unit CIWV (W.Kg$^{-1}$) | TOA | 0.3 | 1.1 | 1.4 |
| | Surface | -0.4 | 1.6 | 1.2 |
| | Convergence | 0.87 | -0.56 | 0.31 |
| per one AOD $SD^*$ (W.m$^{-2}$) | TOA | -1.4 | 8.0 | 6.6 |
| | 500hPa | -6.2 | 10.6 | 4.4 |
| | 700hPa | -14.8 | 11.6 | -3.2 |
| | Surface | -27.0 | 15.8 | -11.3 |
| | Convergence | 25.7 | -7.8 | 17.9 |
| per one CIWV $SD^*$ (W.Kg$^{-1}$) | TOA | 1.7 | 5.8 | 7.5 |
| | 500hPa | -0.4 | 9.3 | 8.9 |

| | | | |
|---|---|---|---|
| 700hPa | -1.6 | 9.4 | 7.9 |
| Surface | -2.4 | 8.3 | 5.9 |
| Convergence | 4.0 | -2.8 | 1.3 |

**Figures**

[Figure]

Figure

[Figure]

**Figure 2**

[Figure]

**Figure 3**

[Figure]

**Figure 4**

[Figure]

**Figure 5**

[Figure]

**Figure 6**

[Figure]

**Figure 75**

[Figure]

[Figure]

Figure 7

[Figure]

**Figure 8**

[Figure]

**Figure 9**

[Figure]

**Figure 10**

[Figure]

[Figure]

Figure 11

[Figure]

[Figure]

**Figure 13**

**Figure 12**

[Figure]

**Figure 14**

[Figure]

[Figure]

[Figure]

[Figure]

[Figure]

**Figure 16**

**Figure 17**

**'The summertime Saharan heat low: Sensitivity of the radiation budget and atmospheric heating to water vapour and dust aerosol' by Netsanet K. Alamirew et al**

The comments and suggestions made by all referees and reviewer are useful. We have addressed the comments raised. Our responses and changes (if any) are indicated in the corrected version of the paper. For clarity we put original comment of the reviewer (typed in italic font) followed by our responses to make it easy to follow.

**Response to interactive discussion Short Comment (SC) from C. Lavaysse**

**Major Comment a.**

1. *Section 3 is not clear. Quite complicated to understand all the configurations and the conclusions drawn from these results on the choice of certain parameters. Finally choices are not really justified and I am not sure if it is necessary to provide all the information. I would recommend to simplify this section and to put some results in the supplementary material.*

   **Response**

   Part of section 3 has been moved to the supplementary material (Section S2). This includes all the model configuration analysis. Accordingly, Section 3 now describes the data and the design of the hypothesis testing experiments and Section 4 focuses only on the results of those experiments.

   **Changes Made**

   We have reorganized section 2 and 3 into a more clear structure. The new structure of the whole paper is as follows.

   Section 1. Introduction
   Section 2. Description of RT model
   Section 3. Data and method
       3.1. Observed top of atmosphere and surface radiation measurements
       3.2. Atmospheric profile and surface characteristics
       3.3. Dust properties and extinction profile
       3.4. RT model Experiments
   Section 4. Results and discussions.
       4.1. RT model validation
       4.2. The radiative flux and heating effects of dust and water vapour
           4.2.1. Dust
           4.2.2. Water vapour
           4.2.3. The relative effects of dust versus water vapour
   Section 5. Summary and Conclusions

Original draft Page 6:L19-40, Page 7:L1-3, Page 7:L28-37 moved to supplementary material (section S2). See also minor comment #3.

45   *2. In this section I also found some parts not clear: p5 l5-12; it is quite weird to compare observations assimilated with model dataset? The authors do not explain the remaining errors. Is it due to the assimilation procedure?*

**Response**

We are pointing the fact that despite assimilation of the radiosonde data there remain biases in the reanalysis. Fennec was a short-term experiment and since then there remains only one radiosonde station for the
50   whole Sahara. As such, the reanalysis errors we derive are almost certainly much lower than those typical of the rest of the Sahara. We also now cite the errors estimated from Garcia-Carreras who compared radiosonde data to a forecast model first guess (independent of assimilation)

The magnitude of errors are different among the different reanalysis products. The possible reasons for the remaining error between observation and reanalysis products could be due to differences in models core
55   dynamics and in assimilation procedures.

**Changes Made**

Corrected draft Page 4. L36-38. A statement added suggesting the possible reasons for differences in error among
60   reanalyses.

**Major Comment b.**

*1. Section 4 is too descriptive with too much information that are not necessarily significant or important to the
65   conclusions of this study. This is particularly true p9 and 10. I strongly recommend to reduce this part to the most important results and to put the others results into an annex.*

**Response**

70   Part of section 4 has been moved to the supplementary Material (section S3), specifically sections describing the sensitivity experiments towards the model optimum configuration, as we agree these are not the key significant results.

We choose to retain some of the results originally presented in pages 9-10 because we feel it is important to demonstrate that the simulated quantities of top of atmosphere radiation budgets are within the observational
75   uncertainties. To give sense of results in subsequent sections, it is necessary to have a feeling of the surface and TOA radiative budget under the mean state.

**Changes Made**

80   Original draft page 8:L30-33, page 9:L3-8 moved to supplementary material (section S3)

*2. The summary of the subsection 4.1 is too speculative. How the authors can conclude the simulated flux errors of the optimal configuration are comparable to the observational uncertainties? What does 'acceptable' mean?*

**Response**

Given that we do not have accurate data for all the input required to run the RT model, it is not unexpected to get some uncertainty in our results. However we have chosen the inputs in such a way that the calculated flux are as close as possible to observation. This is what we mean by an 'optimum' model configuration. The optimum configuration is deemed to be 'acceptable' because the model error in top of atmosphere fluxes (perhaps the single most important quantity) with respect to observations is within the uncertainty in the observational estimates of those quantities. Model estimates lying within observation range is a commonly used indicator of acceptable model performance. Thus we suggested the RT model is configured to produce acceptable results and thus can be used for further experiments.

**Major Comment c.**

1. *Some conclusions are too speculative. The authors conclude about the impacts of the dust aerosols and water vapor on the SHL but, in that study, only June 2011 is used. The SHL is the most important from end of June to mid of September (when it is installed in its Saharan location). Even if the authors used only one month (June), they have to characterize this specific year to the climatology (in term of dust, humidity, large scale forcings). This point concerns the title ('summertime' is not appropriate), the conclusions (p15 l8-10), and the abstract.*

**Response**

We agree that the period of study does not coincide with the peak of the summer season when the SHL is established in its northernmost position. However, we are limited by the period of the Fennec field campaign whose data underpin our analysis. Accordingly we have changed all references to 'summertime' to 'early summer'. In addition, in Section 3.2 we note that during our study period of June 2011 the SHL underwent a rapid transition from a 'maritime phase' to a 'heat low' phase. As such our analysis actually covers the transition period and SH states characteristic of both early and high summer. We have now amended this section to include an analysis of the conditions during June 2011 with respect to the mean conditions during June.

**Changes Made**

References to summer changed to summertime.
Figure 1 changed to show position of SHL in June, 2011.
Corrected draft Page 16:L14-20. A paragraph added

2. *Also the discussion on the impacts on the SHL pulsations should be carefully discussed since the authors do not analyze the contribution of the large scale temperature advections and they never show the real position of the SHL in June 2011 (in June, the SHL is migrating to the north with a large spatial variability).*

**Response**

Real position of SHL in June is shown in Fig 1.

The comments on our reference to variability in SHL specifically the 'pulsating' of SHL intensity and the potential role of dust and water vapour feedbacks in this process is also raised by anonymous referee #1. We do feel it is important in this paper to relate the radiative heating rates derived from our RT simulations to the behaviour of the SHL, but of course recognise that the full dynamical response requires an analysis of advective heating. As such in the original paper p16 para 1 we note that radiative heating is of 'comparable magnitude' to published estimates of advective cooling from comparable monsoon surge type events. In this way we make only a broad inference about the net effects of advective and radiative terms on the SHL. We have now changed the text slightly to emphasise the speculative nature of this inference.

**Changes Made**

Corrected draft Page 15:L26-28. Additional statement included.
'

3. *Finally at climatological scale, the authors should pay attention to the climatological evolution of the dust that tends to reduce (p15 l16)*

**Response**

Our comment in the original draft page 15:L16 concerns other analysis which implicate long term trends in SHL temperature to that in WV, but do not include dust in their analyses. We simply aimed to point out that this should not be neglected. Our paper is not concerned with resolving long term trends in dust over the SHL so we do not include plots of long term satellite derived AOD over the SHL.

**Major Comment d.**

1. *Some figures are not readable*

**Response**

Unreadable figures corrected.

**Minor Comments**

1. *P2 l11 the authors should mention this reference: Lavaysse, C., Flamant, C., Evan, A. et al. Clim Dyn (2016) 47: 3479. doi:10.1007/s00382-015-2847-z*
   **Response:** Reference included, P2:L11 and reference section page 18: L32.
2. *P6 l4; the two phases mentioned are not so clear.*
   **Response:** These two phases are previously stated on original draft page 4:L40 and page 5:L1
3. *P6 l19: title of subsection 3.2 not clear, please rephrase*
   **Response:** changed to 'RT sensitivity experiments to choice of inputs', now moved to supplementary material.

4. *P6 l24: optimal to what?*

**Response:** Optimal configuration means model configured to produce results closest to observations.

5. *P6 l37-38; how do the authors conclude the Ceres measurements are uncertain and that explain the large RMSE? The term RMSE refers to a reference (usually observations) that are considered as the correct value. Here, I do not understand what is the reference and how they can conclude that. Please clarify. Also the term RMSD (difference) should be more appropriate.*

**Response:** We do agree with the reviewer's comment that RMSD is comparison of modeled versus observation. From the data we have CERES is considered correct, despite its limitations as with any observation, can be used to measure the error modelled variables.

**Changes Made:** RMSE changed to RMSD in all occasions.

6. *P6 l39-40: the authors provide some results without explanations, what are these results (mean =...) and please clarify the conclusions/interest of this point?*

**Response:** Rephrased, point of interest described in section 5

7. *P7 subsection 3.2.2 I recommend to put the first part of the paragraph in the introduction section and the result in supplementary material.*

**Response:** Some of the information and results on optical properties of dust is now moved to section S1 of supplementary material.

8. *P8 l1: Section 4.1 is correct?*

**Response:** Corrected

9. *P8 l11: Is it necessary to use this acronym?*

**Response:** Acronym definitions summarized in table 2. To be consistent throughout the paper, we found it necessary to use acronym.

10. *P8 l27: Section 3.1 is correct?*

**Response:** corrected, for the details look at response to Major comment a.

11. *P11 l7-8: longwave and shortwave are equal*

**Response:** TOA SW DRE of dust is small, whereas LW has a net warming effect at TOA(less LW escaping out of atmosphere due to dust.)

12. *P12 l36-37: The SHL is measured in between 925 and 700hPa, not at the surface. Do the authors conclude there is a cooling of the SHL intensity due to the water vapor?*

**Response:** Here we are discussing the immediate radiative effect of dust and water vapour. But the net effect may not be cooling as the feedback resulting from surface warming in the LW and thus more sensible heat flux could result in net warming of the atmosphere which needs further investigation using regional climate models that include the feedback processes.

13. *Figures : For all the figures, please add the caption under the figures*

**Response:** All changes are made to the figures according to the given recommendation.

**Response to Referee Comment (RC) from Anonymous Referee #1**

1. *This paper used field experiment data at BBM in southern Algeria from June 2011 and a radiative transfer model to calculate the effects of dust and water vapor on radiation budget both at the surface and the TOA in order to understand the radiative processes within the SHL during summer. Generally, the manuscript is straightforward*

*and well organized. However my main concern is that some of the input data for the RT model may cause large*
210     *uncertainties that are helpless to fill the research gaps as the authors mentioned in the introduction.*

**Response**

We fully recognise the challenge of adequately constraining the input data to the RT model in this
215     region, where observations are sparse and as a results reanalyses models have limited assimilation of
observations. This is indeed a challenge and one which the Fennec project set out to address. In using Fennec
data we therefore utilise the best available data for our RT simulations. Moreover, we undertake a very
comprehensive analysis of the sensitivity of radiative heating to uncertainties in those input field not directly
measured during Fennec. Indeed reviewer 1 felt that this model configuration section was too comprehensive to
220     be included in the main paper! So we believe we have addressed the issue of data input uncertainty as thorough
and comprehensive manner as could be reasonably expected. This is now included in the supplementary material
section so as not to distract from the core hypotheses the paper sets out the test.

2.  *For example, dust can absorb thermal infrared radiation, the night time AOD estimated from the nephelometer,*
225     *which measures aerosol extinction coefficient near the surface, could induce a large error without an accurate*
*aerosol extinction profile.*

**Response**

230     Lack of complete input data is one of the challenges in the study of radiative effect of aerosols. Because
of this, there is always assumptions or approximations to overcome the arising difficulties. Using surface
nephelometer measurements to estimate night time AOD will not significantly affect our result. This is because
there is only LW forcing at night which is in general smaller compared with SW forcing. Besides researchers
practically use uniform dust extinction profile across the boundary layer as the difference in forcing results
235     compared with the actual extinction profile is not small. [Liao and Seinfeld 1998, Osipov et al., 2015,]
We have also confirmed this through a sensitivity experiment to test the difference in LW radiative flux
and heating rate when we use different daytime and nighttime extinction profile. We find a small difference less
than 3 W.m$^{-2}$ both at the surface and TOA. The atmospheric heating rates do not change significantly when
different extinction profiles are used for day and night except small difference in the lower levels by less than
240     0.20 K day$^{-1}$. We conclude in general that this will not affect what we wanted to show and hence the overall
result of the paper.

3.  *Reanalysis data generally has poor representations of clouds and their properties. However, the authors selected*
*clouds properties from the reanalysis. These could directly affect the reliability of the model results.*
245

**Response**

This was also our concern at the beginning of this research work as we understand the limitations of
cloud representations in models. We could have undertaken the RT experiments only in clear sky mode as many
250     other authors choose to do. We do include clear sky only experiments but we complement these with all sky

experiments to provide a more thorough and comprehensive analysis, from which we compare observations of TOA fluxes in which cloud screening is problematic. Our all sky RT experiments use what we feel is the best available 3-D information on cloud, that comes from the reanalysis models. Alternative cloud profiles for RT models simulations is not available. It is totally expected that our results will bring error due to cloud under (or mis) representation. We discuss this on Page 9: L14-20 of corrected draft and page 3:L1-5, L14-16 of supplementary material. However, we stand by our analysis not least because comparison of the errors in the all sky vs clear sky RT results actually provide some first order indication of the error on radiative budget due to underestimated cloud in reanalysis dataset. We have included a clearer and more explicit caveat regarding the limitations of the cloud fields in our experiments and note the need for further work in this area.

**Changes Made**

Page 3:L14-16 of supplementary material.

4. *Sections 2 and 3 are a bit long. I would recommend to combine and simplify this part.*

**Response**

This part has been restructured in a more clear way (please refer to the comment of reviewer 1, reviewer #1 Major Comment a #1.)

**Changes Made**

Refer to the response of reviewer #1 Major Comment a #1 for the simplified layout of the paper.

5. *What the authors concluded cannot be totally supported only from the radiative forcing and heating rate calculations.*

**Response**

Reviewer #1 also raised this comment. Please refer the responses made to reviewer #1, Major Comment C #2

6. *The manuscript also need a thorough editing. Some typos and confusing expression make the text difficult to follow at times.*

**Response**

Manuscript thoroughly read and corrections made to typos.

**Response to Referee Comment (RC) from Anonymous Referee #2**

**Major Comments**

1. *Error Analysis: The authors spend a good bit of time estimating uncertainty in their modeled fluxes via comparison to satellite retrieved fluxes. However, when it comes to the data analysis, these uncertainties are not taken into consideration. I think it's great that the authors have a handle on the RT model errors, but I think it would be far more useful to carry those uncertainties throughout the entirety of Section 4. Doing so would make the paper and results much stronger and would afford the community opportunity to make a more precise comparison between yours and future dust forcing estimates.*

**Response**

We agree the importance of including error analysis despite we have reduced the uncertainty using sensitivity experiments. This is addressed qualitatively to some extent in section 4, i.e. error associated with the uncertainties in the input.

**Changes made**

Additional information quantitatively expressing the error in flux calculation associated with uncertainties in some of the input data is provided. Page 8 L25-28 and L37-39.

2. *Radiative Transfer Model. To generate the mie coefficients the authors use two different size distributions (Dubovik and Ryder) but the same index of refraction. However, what's the source of the refractive index? The authors conclude that the Dubovik size distribution is more representative of the actual size distribution based on a comparison of the model and observed/retrieved fluxes. However, it is completely possible that the index of refraction used here also biased. For example, it's possible that the Ryder distribution is correct but doesn't produce enough SW dust forcing because the MEC is too low at the appropriate size parameter, thus the forcing in the SWE for Dubovik would better match observations because it's biased towards smaller particles. At any rate, my only point is that you have two degrees of freedom and you can't say conclusively that one size distribution is more representative than another one b/c the index of refraction isn't constrained.*

**Response**

We agree that the refractive index may cause uncertainty in the flux calculations especially in the SW absorption. It is also interesting to test the sensitivity of radiative flux to refractive index. In general for a given size distribution of dust, when refractive index is increased net SW heating will increase and net LW cooling will increase to a lesser extent. This however is a complicated function depending on the surface albedo and cloud. (Liao et al., 1998). Here we used recent measurements for dust refractive index over the Sahara (Ryder et al., 2013) which is function of the composition of dust particles, independent of the size distribution. It could be possible that if we reduce the refractive index, the SW heating will reduce in Ryder distribution, which is the biggest discrepancy compared with satellite measurement. But we haven't made sensitivity test as we have measured refractive index.

3. *RT Model: The authors state that the vertical profile of the dust mass mixing ratio is adjusted so that for a given*

335 *MEC the AOD matches observations. Is the profile linearly scaled by a single value to match the observations? Is a single coefficient derived for all cases or is this done independently for each RT simulation?*

**Response**

340 To be clearer, first an average extinction profile is derived from CALLIOP and this profile is used to derive the extinction profile at each time step, i.e. the average profile is adjusted to match the measured AOD from AERONET. So to answer the question, for each RT calculations independent extinction profile is derived.

4. *Flux comparisons: It the text it is not clear if the flux comparisons are performed in a robust manner. For*
345 *example, why are monthly mean fluxes from CERES compared to the observations and output from the model? The proper way to conduct the comparison with CERES would be to access the daily nighttime and daytime data and then sub sample the observations/RT model output/GERB retrievals in order to conduct an apples-to-apples comparison. The authors acknowledge this (Page 9 line 35) so it's puzzling why a more thorough analysis wasn't performed. This approach includes the task of making comparisons to the reanalysis data (again, authors note*
350 *that interpolating MERRA surface temperature may be biasing the flux comparisons). Furthermore, more insight would likely be gained by comparing the clear-sky fluxes only, since cloud forcing is not important to the study.*

**Response**

355 An important aspect of this study that needs to be noted is it is intended to provide season (one month) study of the radiative budget and sensitivities to water vapour and dust variability over the Saharan heat low. In order to do so we have used the best available input dataset through sensitivity experiments. It is useful to carry out comparison of the radiative flux at the time steps of CERES data (which is twice per day) as the referee suggested. We have actually made comparison of RT model outputs with CERES data with the respective time
360 step to derive RMSE. This is presented on page 9 line 21(corrected draft). This will give us a good picture of the uncertainties of model simulations. However further comparisons using average of two time steps per day will not enable us to achieve the target we put at the outset.

To compare simulated flux with observation, GERB data is used. Further reanalysis data is also used
365 which is available daily and thus used the same days as the RT model simulation days. CERES data is not used to compare simulated flux except for sensitivity experiments and estimate cloud DRE. We understand that using month mean CERES clear sky and all sky flux will bring some error but it will give us first order estimate of cloud DRE over the region. This will help emphasize need to improve the error on the radiative budget due to underestimated cloud in reanalysis dataset despite the challenges in making these comparisons.

370

5. *Flux comparisons: Tables and Figures. There are too many tables and the main figure (9) for this section is not particularly useful. Firstly, the tables are cumbersome and don't communicate the main results well (for example, color could be used to indicate if RT model output or reanalysis output is biased high or low in comparison to surface obs or satellite retrievals. In addition, the flux comparison Fig 9 are tough to interpret*
375 *because the annual cycle is included. A better way to do this is to have one plot comparing the mean annual cycles, and another comparing the anomalies.*

**Response**

380    We agree to remove Table 5 since the information on this table is also found in Tables 2 and 3(corrected draft). An additional table is moved to the supplementary material.

Colours included on the wDwC results in tables 3 and 4(corrected draft Tables 2 and 3) red indicating

385    model results overestimated and blue indicating model results underestimated compared with observation.

Some of the figures were corrected based on referee #1 and reviewer's comments. Figure 9(also Figure 6) is corrected and it is easier to read. We therefore keep it as it is. But have also made additional plot using anomalies but we put it in the supplementary document. See also page 9 L18-20.

390    **Changes made**

Table 5 removed

Colours used on column 6 of table 2 and table 3

Additional figure included in supplementary material page 3, figure S2

395

6.  *Forcing efficiencies for dust and CIWV should also include the 95% confident interval from linear regressions.*

**Response**

400

We agree that the regressions should be expressed to 95% confidence level. All the regression results are expressed within the 95% confidence interval.

**Changes made**

405

These are included in section 4.2.1(page 10-11) and section 4.2.2(page 12) on the corrected draft.

7.  *Figure 12 and 16 are not interesting. Consider including observations here as well (at least for TOA). BTW - CERES produces surface flux products. These could be folded into the analysis as well.*

410    **Response**

Here the plots are made using daily averaged variations in dust AOD or water vapour. That is dust AOD (and CIWV) is increased linearly in each RT run. This is a theoretical work designed to investigate the sensitivity of dust and water vapour on the radiation budget. There is no such observational data, at least at one particular

415    point which is the observational data we used here. This can be tested for a number of grid points of Satellite observation to see sensitivity of radiation to AOD variation (e.g Young et al., 2009). However this is not the objective of this study and thus it is not included.

420    8. *Figure 17 is impossible to read/interpret, and I don't even wear glasses (yet)! Please consider a more simple and straightforward way to describe the vertical sensitivities. A good rule-of-thumb would be to only include in the plot information that you actually describe in the text.*

**Response**

425

Figure made easier to read. Additional explanation regarding the figures provided

**Changes made**

430    Now we put the two panels of figure 17 as independent plots, Figure 16 and Figure 17 in the corrected draft. Statement added on page 13, L18-19.

**Minor Comments**

435    1. *Individual panels of the figures should be labeled as a,b,c,...*

**Response**

All figures prepared accordingly

440

2. *Figure 5: This figure is not very useful in terms of understanding the relationship between the AODs and IWV. Can you please just replace with one or two scatter plots?*

**Response**

445

We used SEVIRI AOD to show that there are cases where AOD is missed in AERONET which we suggest to be due to confusing dust with cloud. This we believe is important to show there are cases where dust might be missed in AERONET. We have complemented this using nephelometer measurements.

450    3. *Figure 6. If the authors removed the diurnal cycle from this plot we'd have an easier time interpreting the magnitude of the biases. As it is presented here, the magnitude of the differences are small relative to the magnitude of the diurnal temperature changes, making it difficult to interpret the results.*

**Response**

455

Figure 6 is now made easier to read and thus we keep it as it is. In addition we put the anomalies in the supplementary material. Additional information included in the supplementary material SP2 L:21-22

4. *Page 9, Line 2: You write "Dubovik Optical Properties" do you mean optical properties generated using the size distribution from Dubovik and the index of refraction that you've been using up to now (that hasn't been referenced)? It's just not clear.*

**Response**

Restated. Now on page 7 line 10 and 14.

Refractive index used comes from measurement. It is now made clear, Citation included, page 5 line 36

5. *Page 12, Paragraph starting on line 28: The finding that IWV and AOD contribute approximately equally to variance in the radiative budget is by far the most interesting (and new) finding reported in the paper. Why not take a little more space to flesh this out a bit? And please include the uncertainty estimates.*

**Response**

We agree this is an important point. Additional statement highlighting the significance of dust on controlling the radiative budget is included. Page 13 Line 4 of corrected draft.